# Tachyonic and parametric instabilities
# in an extended bosonic Josephson junction

**Laura Batini[1][*], Sebastian Erne[2][†], Jörg Schmiedmayer[2], and Jürgen Berges[1]**

**1** Institut für Theoretische Physik, Universität Heidelberg, Philosophenweg 16, 69120 Heidelberg, Germany
**2** Vienna Center for Quantum Science and Technology, Atominstitut, TU Wien, Stadionallee 2, 1020 Vienna, Austria

[*] batini@thphys.uni-heidelberg.de , † sebastian.erne@tuwien.ac.at

## Abstract

We study the decay of phase coherence in an extended bosonic Josephson junction realized via two tunnel-coupled Bose-Einstein condensates. Specifically, we focus on the $\pi$-trapped state of large population and phase imbalance, which, similar to the breakdown of macroscopic quantum self-trapping, becomes dynamically unstable due to the amplification of quantum fluctuations. We analytically identify early tachyonic and parametric instabilities connected to the excitation of atom pairs from the condensate to higher momentum modes along the extended direction. Furthermore, we perform Truncated Wigner numerical simulations to observe the build-up of non-linearities at later times and explore realistic experimental parameters.

# 1   Introduction

The Josephson effect based on quantum tunneling through a potential barrier between two superconductors or superfluids [1, 2] is a clear manifestation of macroscopic quantum phase coherence, which is a central resource [3] for fundamental studies and technological applications. Ultracold atomic gases provide a particularly rich platform for studying the physics of Josephson tunneling [4–6]. Not only do they provide exceptional control over system parameters and advanced imaging techniques to monitor the details of equilibrium and out-of-equilibrium properties, most interestingly, they provide an additional richness by the ability to control the interaction within the superfluid.

Even the simplest zero-dimensional (0D) Josephson junction exhibits remarkably rich dynamics and strongly depends on whether the atom-atom interaction energy or the tunneling energy dominates. The first theoretical prediction of coherent atom exchange between two weakly linked atomic Bose-Einstein condensates (BEC) was proposed by Javanainen [4], Milburn et al. [5] then formulated the first explicit two-mode Hamiltonian for this system. The resulting Josephson-like dynamics in *weakly* coupled condensates was then described in detail for both DC and AC Josephson regimes in [6].

A systematic classification of dynamical regimes within the two-mode approximation taking into account the atom-atom interactions was worked out in [7,8]. Depending on the relation between the difference of the interaction energy between the wells ($\Delta E_{int}$) and the tunneling energy ($\hbar J$) one observes either regular full Josephson oscillations around a zero average population difference ($\Delta E_{int} < \hbar J$), or an unbalanced state with nonzero average population difference and only a small fraction of atoms tunnel between the two wires ($\Delta E_{int} > \hbar J$). In the latter, the relative phase between the two condensates can either monotonically increase in time, corresponding to a running phase, referred to as macroscopic quantum self-

trapping (MQST), or be trapped, oscillating close to $\pi$, referred to as the $\pi$-trapped state, or $\pi$-oscillation [7].

To study relaxation and decay of Josephson oscillations in these bosonic Josephson junctions (BJJ), one has to extend beyond the two-mode approach. [9] investigated a multimode expansion demonstrating that $\pi$-states and self-trapped configurations can dynamically destabilize through internal coupling, even in the absence of dissipation. Similarly, coupling the two-mode Josephson oscillations to a bath of higher quasiparticle modes leads to eventual relaxation of the BJJ [10].

Albiez et al. [11, 12] first observed Josephson oscillations and macroscopic self-trapping in 0D double-well BECs. Cataliotti et al. [13] realized a one-dimensional array of bosonic Josephson junctions using an optical lattice and observed coherent atomic current oscillations, enabling the study of phase-coherent dynamics in extended Bose–Einstein condensate systems. Levy et al. [14] later demonstrated both the a.c. and d.c. Josephson effects in such condensates further enriching our understanding of these phenomena. The physics of BJJ was also studied in exciton-polariton condensates [15].

Moving from zero-dimensional to *extended bosonic Josephson junctions* (eBJJ) introduces qualitatively new phenomena. This system hosts a number of different instabilities, enabled by the extended spatial direction, which are absent in 0D. Previous theoretical studies include the pure Josephson regime [16–19], connecting the eBJJ to the Sine-Gordon (SG) model [20], the formation of oscillons (long-lived, spatially localised excitations of the SG-model) [21, 22] or parametric instabilities induced by explicit modulation of the tunneling coupling [23]. Phenomenological damping has also been incorporated to model relaxation toward equilibrium [24]. In the macroscopic quantum self-trapping (MQST) regime, a dynamical instability is predicted to emit correlated quasiparticle pairs, consistent with truncated Wigner simulations [25].

Experimentally, a promising approach to an extended-junction geometry employs, starting from a one-dimensional array of bosonic Josephson junctions [13], two parallel, weakly coupled elongated quasi one dimensional BECs [26–31], which allows direct measurement of density and phase fluctuations along the junction and thereby characterize the physics in full detail [32–34]. Decay of both the Josephson and MQST regimes has been observed experimentally in coupled condensates [35, 36], although precise comparison with theory is limited by harmonic confinement [36, 37]. To date, the only direct realization of a metastable $\pi$-state was in a superfluid $^3$He weak link [38], and its observation in extended cold-atom junctions remains an open challenge.

In this work, we present the first detailed theoretical analysis of $\pi$-trapped state instabilities in extended bosonic Josephson junctions. We combine analytic predictions for tachyonic and parametric instability bands with Truncated Wigner simulations that capture nonlinear secondary processes. Importantly, we identify concrete experimental signatures — from imbalance dynamics to momentum-resolved twin-beam correlations — that provide a clear pathway toward observing these instabilities with present-day cold-atom technology.

First, we recapitulate the spatially uniform mean-field dynamics, which is stable. Next, we analyze small, spatially dependent fluctuations. Specifically, first we expand the dynamics up to first order in space-time dependent perturbations. We find that bands of momentum modes with nonzero wave-vector are exponentially growing and quantify the corresponding growth rate. One can distinguish primary instabilities where the growth of characteristic modes is shown to be of tachyonic and parametric resonance origin [39–41]. To go beyond the linearized analysis, we compare with the numerical results obtained from simulations using a truncated Wigner Approximation (TWA) [42]. This allows us to identify secondary instabilities [39, 40], which represent the even faster non-linear growth of modes triggered by the primary instabilities [39, 40]. Our study provides the basis for quantitative comparisons with

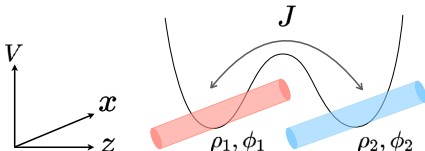

Figure 1: Sketch of the experimental setup under study, consisting of two elongated condensates (left 1 and right 2). The BECs are characterized by an atomic density $\rho_{1,2}$ and a phase $\phi_{1,2}$. The BECs are coupled by a single-particle tunnel interaction $J$.

future experimental implementations.

This paper is organized as follows. In Sec. 2, we introduce the theoretical model and the dynamical regime of the $\pi$-mode. In Sec. 3, we analyze the early dynamics of this many-body system, focusing on the instability chart for the primary instabilities. In the following section 4, we demonstrate numerically the existence of secondaries being present in this system and discuss their generation. Then, we also relate to possible experimental realizations in Sec. 5. Finally, Sec. 6 concludes and gives an outlook. The appendices provide a brief review of dynamical instabilities, the mean-field picture, and information on the implementation of the numerical classical-statistical simulations.

## 2 Tunnel-coupled Bose condensates

We consider a system of two weakly tunnel-coupled elongated BECs, which are trapped in a double-well potential [43, 44]. This section summarizes the theoretical model and derives the relevant equations for the $\pi$-state using a density-phase representation. For a more detailed discussion on the experimental implementation and feasibility, we refer to Sec. 5.

### 2.1 Model and equations of motion

The system under consideration comprises two quasi-1D Bose gases, each trapped in one well of a double-well potential, as depicted in Fig. 1. They are described by the following Hamiltonian [16]

$$\hat{H} = \int_0^L dx \left[ -\frac{\hbar^2}{2m} \sum_{j=1}^2 \hat{\Psi}_j^\dagger \partial_x^2 \hat{\Psi}_j + \frac{g}{2} \sum_{j=1}^2 :\left(\hat{\Psi}_j^\dagger \hat{\Psi}_j\right)^2: \right] - \hbar J \int_0^L dx \left( \hat{\Psi}_1^\dagger \hat{\Psi}_2 + \hat{\Psi}_2^\dagger \hat{\Psi}_1 \right). \tag{1}$$

Here $L$ is the system size, $m$ is the atom mass, and $\hat{\Psi}_j(t,x)$ $(j=1,2)$ represents the bosonic field operator, which obeys the canonical equal-time commutation relations

$$\begin{aligned}
\left[\hat{\Psi}_i(t,x), \hat{\Psi}_j(t,x')\right] &= \left[\hat{\Psi}_i^\dagger(t,x), \hat{\Psi}_j^\dagger(t,x')\right] = 0, \\
\left[\hat{\Psi}_i(t,x), \hat{\Psi}_j^\dagger(t,x')\right] &= \hbar \delta_{ij} \delta(x-x'),
\end{aligned} \tag{2}$$

with $i,j = 1,2$. The Hamiltonian consists of kinetic energy, contact interactions of strength $g$, and tunneling of amplitude $J$ between the two condensates. Due to the spatial separation of the two condensates, intra-species interactions are considered to be negligible.

Throughout we adopt a *spatially resolved two-mode* description that retains the full $x$-dependence of the fields $\hat{\Psi}_{1,2}(x,t)$. In the homogeneous mean-field limit (Subsec. 2.3), i.e.,

for spatially uniform fields, we recover the Smerzi two-mode approximation [45]; beyond that limit (Sec. 3), spatial fluctuations couple to the zero mode, leading to instabilities absent in the standard two-mode model.

## 2.2 Equations of motion

The coupled evolution equations for the 2-component BEC follow from the Heisenberg equations of motion derived from the Hamiltonian in Eq. (1),

$$
i\hbar \frac{\partial}{\partial t} \hat{\Psi}_1 = -[\hat{H}, \hat{\Psi}_1] = \left( -\frac{\hbar^2 \partial_x^2}{2m} + g|\hat{\Psi}_1|^2 \right) \hat{\Psi}_1 - \hbar J \hat{\Psi}_2,
\tag{3}
$$

and similarly for $\hat{\Psi}_2$ by replacing $\hat{\Psi}_1 \leftrightarrow \hat{\Psi}_2$ in Eq. (3).

In order to study the quantum dynamics approximately, we adopt a semi-classical perspective. Specifically, we consider the Gross-Pitaevskii equations (GPEs) of motion that are obtained after replacing the Bose field operator in Eq. (3) by a corresponding classical field with $\hat{\Psi}_j(t, x) \to \Psi_j(t, x)$. In the semi-classical description, the statistical averages of the classical fields have to fulfill the same initial conditions at a given time $t_0$ as the quantum fields for the quantum expectation values of their means, $\langle \hat{\Psi}_j(t_0, x) \rangle$, and their fluctuations encoded in two-point correlations, $\langle \hat{\Psi}_j(t_0, x) \hat{\Psi}_k(t_0, y) \rangle$ or also higher correlations, which are suitably symmetrized to describe the equal initial times. The semi-classical approach is particularly useful to derive the linearized evolution equations for fluctuations below and for the non-linear TWA description of Sec. 4. For more details, we refer to Appendix C.

Furthermore, it is convenient to consider the Madelung representation, where the fields $\Psi_j(t, x)$ for $j = 1, 2$ are expressed in terms of the condensate densities $\rho_j(t, x)$ and phases $\phi_j(t, x)$, such that

$$
\Psi_j = \sqrt{\rho_j} \, e^{i\phi_j} \, .
\tag{4}
$$

In terms of these variables the coupled GPEs become for the densities and phases

$$
\begin{aligned}
\hbar \dot{\rho}_1 &= -\frac{\hbar^2}{m} \left( \rho_1 \partial_x^2 \phi_1 + \partial_x \rho_1 \cdot \partial_x \phi_1 \right) - 2\hbar J \sqrt{\rho_1 \rho_2} \sin(\phi_2 - \phi_1), \\
\hbar \dot{\phi}_1 &= \frac{\hbar^2}{2m} \left( \frac{\partial_x^2 \rho_1}{2\rho_1} - \frac{(\partial_x \rho_1)^2}{4\rho_1^2} - (\partial_x \phi_1)^2 \right) - g\rho_1 + \hbar J \sqrt{\frac{\rho_2}{\rho_1}} \cos(\phi_2 - \phi_1),
\end{aligned}
\tag{5}
$$

respectively and correspondingly for $\rho_2$ and $\phi_2$. Here, $\dot{\rho}_j \equiv \partial \rho_j / \partial t$, and equivalently for $\phi_j$, denotes the derivative with respect to time. It is convenient to define the relative degrees of freedom, $z \equiv (\rho_1 - \rho_2)/(\rho_1 + \rho_2)$, with $-1 < z < 1$, and the relative phase $\phi \equiv \phi_1 - \phi_2$.

## 2.3 Two-mode approximation

We first consider the mean field evolution of a purely homogeneous background field configuration such that $\rho_j(x, t) = \bar{\rho}_j(t), \phi_j(x, t) = \bar{\phi}_j(t)$. In terms of the mean field relative imbalance $\bar{z}$ and phase $\bar{\phi}$ they are given by

$$
\begin{aligned}
\hbar \dot{\bar{z}}(t) &= -2\hbar J \sqrt{1 - \bar{z}^2(t)} \sin \bar{\phi}(t), \\
\hbar \dot{\bar{\phi}}(t) &= \hbar J \left[ \Lambda \bar{z}(t) + \frac{2\bar{z}(t)}{\sqrt{1 - \bar{z}^2(t)}} \cos \bar{\phi}(t) \right],
\end{aligned}
\tag{6}
$$

which are invariant under discrete shift symmetry $\bar{\phi} \to \bar{\phi} + 2\pi l, l \in \mathbb{N}$. We refer to Ref. [7] and Appendix B for details of the derivation. In Eq. (6), the dimensionless parameter $\Lambda$, defined by

$$
\Lambda \equiv \frac{\mu}{\hbar J},
\tag{7}
$$

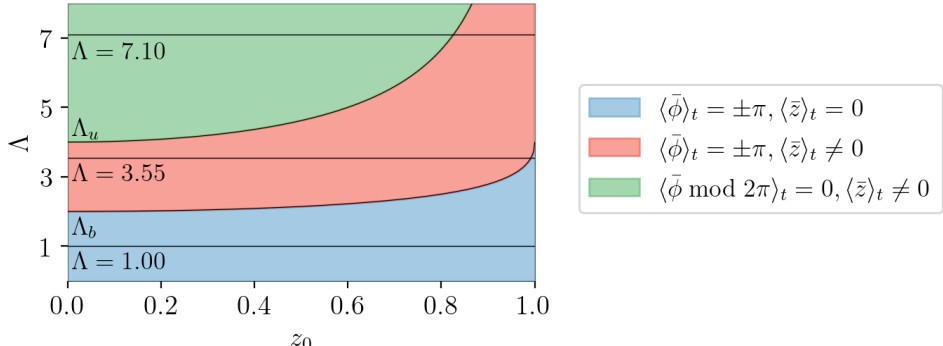

Figure 2: Parameter space $(\Lambda, z_0)$ showing the dynamical regimes delineated by $\Lambda_b$ [Eq. (9)] and $\Lambda_u$ [Eq. (10)], for an initial phase difference of $\phi_0 = \pi$. The horizontal lines represent $\Lambda = \{1.00, 3.55, 7.10\}$.

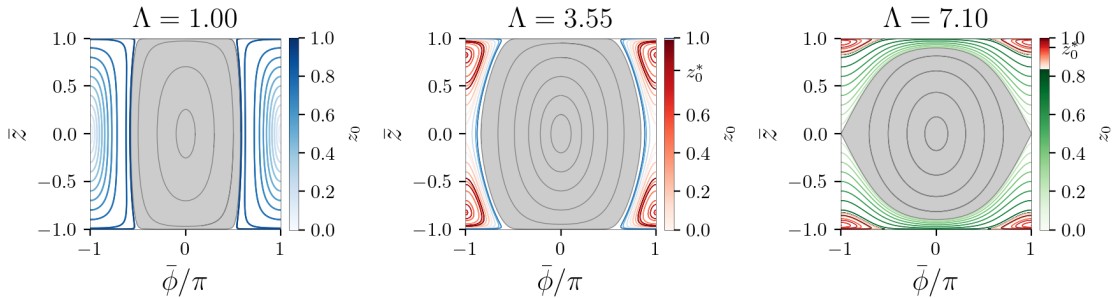

Figure 3: The dynamics of the mean fields $\bar{\phi}$ and $\bar{z}$ corresponding to Fig. 2. For $\Lambda < \Lambda_b(z_0)$, the system exhibits $\pi_0$-oscillations, characterized by $\langle \bar{\phi} \rangle_t = \pi, \langle \bar{z} \rangle_t = 0$ as visible in the left and center panels. For $\Lambda_b(z_0) < \Lambda < \Lambda_u(z_0)$, the system displays $\pi$-oscillations, where $\langle \bar{\phi} \rangle_t = \pi, \langle \bar{z} \rangle_t \neq 0$ as visible in the center and right panels (blue lines). For $\Lambda > \Lambda_u(z_0)$, the system exhibits MQST self-trapped modes, characterized by $\langle \bar{\phi} \rangle_t = 0, \langle \bar{z} \rangle_t \neq 0$ and it is shown in the right plot (green lines). In all plots, the gradient bar indicates the initial condition for $z_0$.

characterizes the ratio between the interatomic interaction and the tunneling energies, $\mu_j = \rho_j g$ is the chemical potential in each well, and $\mu = \mu_1 + \mu_2$ denotes the total chemical potential. In this work, we consider repulsive interatomic interactions, such that $g$ and $\Lambda$ are always positive.

### 2.3.1 Dynamical regimes

The system exhibits a range of different behaviors for various initial conditions. The dynamical modes are fully determined by the initial conditions $(\phi_0, z_0)$ and the dimensionless parameter $\Lambda$. In the following, we focus our discussion on the $\pi$-state dynamics and, therefore, without loss of generality, fix the initial condition for the relative phase to $\phi_0 = \pi$[1]. The different possible dynamical regimes are conveniently represented in parameter space $(\Lambda, z_0)$ [36], which are illustrated in Fig. 2. Let us review the three qualitatively different regimes for a fixed initial condition for $z_0$.

---

[1]It is important to note that the commonly discussed Josephson oscillation regime is excluded due to the initial condition choice of $\phi_0 = \pi$. We have indicated this regime with the gray shaded area in Fig. 3.

- (Only) phase trapping, occurring for $\Lambda < \Lambda_b(z_0)$. The dynamics in this regime presents slightly deformed oscillations of the imbalance, such that $\langle \bar{z} \rangle_t = 0$. The expression $\langle ... \rangle_t$ indicates the temporal average. The relative phase is trapped around $\pi$, i.e., $\langle \bar{\phi} \rangle_t = \pm\pi$. This phenomenon is called $\pi_0$-*oscillations* and is shown in the left panel of Fig. 3.

- Phase *and* density trapping, occurring for $\Lambda \in (\Lambda_b, \Lambda_u)$. This phenomenon is known as the $\pi$-*oscillations*. In this case, $\langle \bar{\phi} \rangle_t = \pm\pi$ and $\langle \bar{z} \rangle_t \neq 0$. As shown in the central and right panels of Fig. 3, the imbalance oscillations are centered around one of the four limiting points $\pm z_0^*$, where the amplitude of the oscillations drops to zero. Analytically, these points are located at [36]

$$(\bar{\phi}, \bar{z}) = \left( \pm\pi, \pm\sqrt{1 - \frac{4}{\Lambda^2}} \right). \tag{8}$$

In order to observe the $\pi$-oscillations, it is essential to carefully choose the initial condition $z_0$ (or $\Lambda$). The lower bound $\Lambda_b$ is determined by the initial imbalance $z_0$. It arises from the requirement that the effective potential for the imbalance (see Appendix B.2 and, in particular, Eq. (B.7) for more details) must be in the symmetry-broken phase to ensure that the imbalance remains trapped at all times. Additionally, the total energy must be negative to prevent the imbalance from surpassing the potential barrier. This energy condition is even more stringent, as discussed in Refs. [25, 45], and gives

$$\Lambda_b = \frac{4\left(1 - \sqrt{1 - z_0^2}\right)}{z_0^2}. \tag{9}$$

The upper bound $\Lambda_u$ amounts to

$$\Lambda_u = \frac{4}{\sqrt{1 - z_0^2}}. \tag{10}$$

This bound results from the condition that the phase remains trapped at all times.

- (Only) imbalance trapping, occurring for $\Lambda > \Lambda_u$. This regime corresponds to *macroscopic quantum self-trapping (MQST)*, characterized by $\langle \bar{z} \rangle_t \neq 0$ and the circular mean of the relative phase given by $\langle \bar{\phi} \mod 2\pi \rangle_t = 0$. The corresponding time evolution of the mean fields is shown in the right panel of Fig. 3.

In this work, unless stated otherwise, we set $\Lambda = 3.55$. This value ensures that, for nearly all initial values of $z_0$, the system remains in the $\pi$-trapped regime [see Fig. 2]. At the same time, it places the system close to the perturbative regime in the coupling ($1/\Lambda \ll 1$).

## 3   Instabilities from linearized fluctuation equations

In this section, we investigate the stability of the mean-field trajectories when transitioning from the 0D bosonic Josephson junction to an extended junction. Here, we focus on the breakdown of the $\pi$-oscillations. In general, the extended direction enables the decay of this state through the excitation of correlated pairs of atoms, energetically prohibited in 0D.

At short times, when the occupation of the non-condensate modes is small, we linearize the equations of motion in the fluctuations around the previously discussed mean fields $\bar{\rho}_j$ and $\bar{\phi}_j$, which allows us to obtain analytical predictions of the instability bands in different regimes.

We characterize the instabilities by their dispersion relations. In particular, the non-zero imaginary components lead to their exponential growth. First, we consider the simple case where the relative phase and the density of the condensates are initialized at a mean-field stationary point. Then, we extend the calculation of the linearized evolution to more general cases, considering the presence of oscillations around the mean-field stable point. In this case, parametric oscillations lead to further instability bands.

## 3.1 Exponential amplification of fluctuations

We write each condensate density $\rho_j$ and phase $\phi_j$ in terms of their homogeneous means $\bar{\rho}_j(t)$ and $\bar{\phi}_j(t)$, with small fluctuations $\delta\rho_j(t,x)$ and $\delta\phi_j(t,x)$ on top as

$$
\begin{aligned}
\rho_j(t,x) &= \bar{\rho}_j(t) + \delta\rho_j(t,x), \\
\phi_j(t,x) &= \bar{\phi}_j(t) + \delta\phi_j(t,x).
\end{aligned}
\tag{11}
$$

Linearizing Eq. (5), we recover to zeroth order in the phase and density fluctuations the mean-field equations corresponding to Eq. (6). To linear order in the fluctuations, the following equations are obtained [46, 47]:

$$
\begin{aligned}
\hbar\delta\dot{\rho}_1 &= -\frac{\hbar^2\bar{\rho}_1}{m}\partial_x^2\delta\phi_1 - \hbar J\sqrt{\bar{\rho}_1\bar{\rho}_2}\sin\bar{\phi}\,(\delta_1+\delta_2) - 2\hbar J\sqrt{\bar{\rho}_1\bar{\rho}_2}\cos\bar{\phi}\,(\delta\phi_2-\delta\phi_1), \\
\hbar\delta\dot{\rho}_2 &= -\frac{\hbar^2\bar{\rho}_2}{m}\partial_x^2\delta\phi_2 + \hbar J\sqrt{\bar{\rho}_1\bar{\rho}_2}\sin\bar{\phi}\,(\delta_1+\delta_2) + 2\hbar J\sqrt{\bar{\rho}_1\bar{\rho}_2}\cos\bar{\phi}\,(\delta\phi_2-\delta\phi_1), \\
\hbar\delta\dot{\phi}_1 &= \frac{\hbar^2}{4m\bar{\rho}_1}\partial_x^2\delta\rho_1 - g\delta\rho_1 + \hbar J\sqrt{\frac{\bar{\rho}_2}{\bar{\rho}_1}}\left(\cos\bar{\phi}\frac{(\delta_2-\delta_1)}{2} - \sin\bar{\phi}\,(\delta\phi_2-\delta\phi_1)\right), \\
\hbar\delta\dot{\phi}_2 &= \frac{\hbar^2}{4m\bar{\rho}_2}\partial_x^2\delta\rho_2 - g\delta\rho_2 - \hbar J\sqrt{\frac{\bar{\rho}_1}{\bar{\rho}_2}}\left(\cos\bar{\phi}\frac{(\delta_2-\delta_1)}{2} + \sin\bar{\phi}\,(\delta\phi_2-\delta\phi_1)\right),
\end{aligned}
\tag{12}
$$

where we define $\delta_j = \delta\rho_j/\bar{\rho}_j$. These equations are valid for linear perturbations around any arbitrary time-dependent homogeneous background. From now on, we focus on the $\pi$-state, which is characterized by closed classical trajectories in $(\phi,z)$ space around one of the non-equilibrium points, see Eq. (8). In this state, both the imbalance and the relative phase oscillate around a non-zero value. For more details on the dynamical regimes and the conditions for the $\pi$ trapping, we refer to the discussion in Appendix B.1.

## 3.2 Tachyonic instability

We specify the linearized equations [Eq. (12)] to the $\pi$-trapped case by setting the fields to their stationary values [Eq. (8)]

$$
\bar{\rho}_j \to \rho_j^*, \quad \bar{\phi} \to \pi,
\tag{13}
$$

and we choose, without loss of generality, that $\rho_1^* > \rho_2^*$.

For the analysis, it is convenient to introduce rescaled time $\tau$ and space $\tilde{x}$ as

$$
\tau \equiv \frac{\mu}{\hbar}t, \quad \tilde{x} \equiv \frac{1}{\xi}x,
\tag{14}
$$

where the healing length $\xi$ is defined as

$$
\xi = \frac{\hbar}{\sqrt{m\mu}}.
\tag{15}
$$

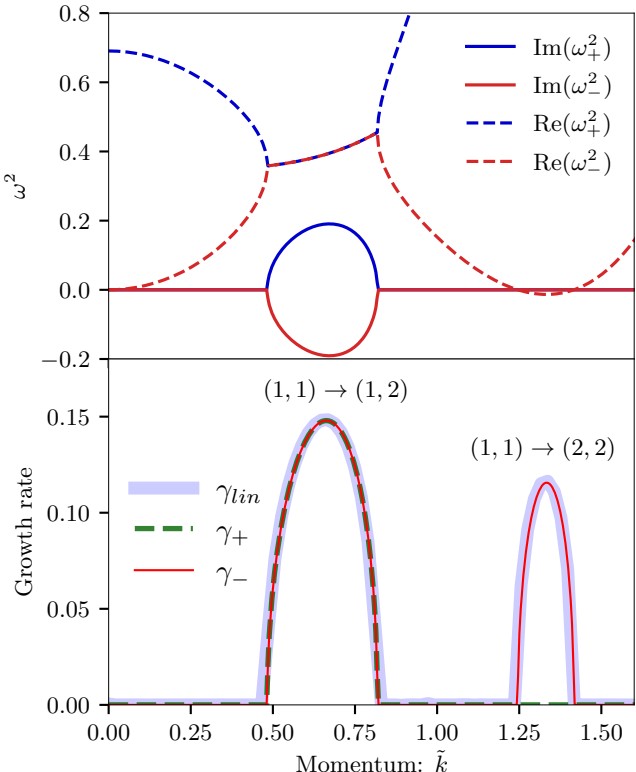

Figure 4: *Top:* Squared dispersion relation $\omega_\pm^2$ [Eq. (17)] as a function of momentum $\tilde{k}$ for $\Lambda = 3.55$. The real parts $\text{Re}(\omega_\pm^2)$ and imaginary parts $\text{Im}(\omega_\pm^2)$ of the squared dispersion relation $\omega_\pm^2$ are represented by solid and dashed lines, respectively. *Bottom:* Growth rates $\gamma_+$ (dashed line) and $\gamma_-$ (thin solid line) corresponding to the imaginary part of $\omega_+$ and $\omega_-$, respectively [from Eq. (17)]. The growth rate $\gamma_{lin}$ (thick solid line) is obtained from solving Eq. (12).

Note that, therefore, frequencies are in units of $\mu/\hbar$ and hence have the same dimension as energies. Note that in the interest of readability, we omit the tilde when there is no risk of confusion. In terms of the rescaled variables, Eq. (12) reads in momentum space

$$\delta\dot\rho_1 = \rho_1^* \tilde{k}^2 \delta\phi_1 + \frac{2}{\Lambda}\sqrt{\rho_1^*\rho_2^*}(\delta\phi_2 - \delta\phi_1),$$

$$\delta\dot\rho_2 = \rho_2^* \tilde{k}^2 \delta\phi_2 - \frac{2}{\Lambda}\sqrt{\rho_1^*\rho_2^*}(\delta\phi_2 - \delta\phi_1),$$

$$\delta\dot\phi_1 = -\left(\frac{\tilde{k}^2}{4\rho_1^*} + 1\right)\delta\rho_1 - \sqrt{\frac{\rho_2^*}{\rho_1^*}}\frac{(\delta_2 - \delta_1)}{2\Lambda}, \tag{16}$$

$$\delta\dot\phi_2 = -\left(\frac{\tilde{k}^2}{4\rho_2^*} + 1\right)\delta\rho_2 + \sqrt{\frac{\rho_1^*}{\rho_2^*}}\frac{(\delta_2 - \delta_1)}{2\Lambda},$$

where now the dot refers to the derivative with respect to $\tau$. One observes from Eq. (16) that the dynamics of $\delta\rho_1, \delta\rho_2, \delta\phi_1$ and $\delta\phi_2$ is coupled to each other. Assuming the solutions have a time dependence $\sim e^{-i\omega(\tilde{k})\tau}$, we find by diagonalization two pairs of eigenfrequencies

$\omega_\pm(\tilde{k})$. Their squared dispersion relations are given by

$$\omega_\pm^2(\tilde{k}) = \frac{X(\tilde{k})}{4\rho_{12}} \pm \frac{\sqrt{\Delta(\tilde{k})}}{2\rho_{12}^2}, \tag{17}$$

where

$$X(k) = 2\Lambda^{-2} - 2\Lambda^{-1}\sqrt{\rho_{12}}\left(4\rho_{12} + k^2\right) + \rho_{12}k^2(2 + k^2), \tag{18}$$

$$\Delta(k) = \Lambda^{-2}\left[-4(\rho_{12})^{3/2} + \Lambda^{-1}\right]^2 - 2\Lambda^{-2}\sqrt{\rho_{12}}\left[\Lambda^{-2} + \sqrt{\rho_{12}}\left((\rho_1^*)^2 - 6\rho_{12} + (\rho_2^*)^2\right)\right]k^2 \\ + \rho_{12}\left[2\Lambda^{-1}\sqrt{\rho_{12}}(\Delta\rho)^2 + \rho_{12}(\Delta\rho)^2 + \Lambda^{-2}\right]k^4, \tag{19}$$

and, for brevity, $\rho_{12} = \rho_1^*\rho_2^*$ and $\Delta\rho = \rho_1 - \rho_2$. The squared eigenvalues $\omega_\pm^2$ are shown in the top plot of Fig. 4 as a function of scaled momentum. The real parts $\text{Re}(\omega_\pm^2)$ and imaginary parts $\text{Im}(\omega_\pm^2)$ are indicated by dashed and solid lines, respectively. From this, one can conclude that the system becomes unstable in two sets of momentum modes for different reasons. The population of these sets of unstable momentum modes grows exponentially. In fact, there is a first range of modes that experience a positive growth rate because $\text{Re}(\omega_-^2) < 0$. This kind of instability is commonly referred to as *tachyonic* instability in a cosmological context [48]. The second region of dynamical instability in the system arises when, in the squared dispersion relation, the square root argument $\Delta(k) < 0$. We note that there are no additional primary unstable regions in the system.

The growth rates $\gamma_+$ and $\gamma_-$, associated with the analytical prediction for the imaginary part of the eigenvalues $\omega_+$ and $\omega_-$, are shown in the bottom plot of Fig. 4 by the dashed and thin solid lines, respectively. The $\gamma_{lin}$ represents the growth rate computed numerically by solving the linearized equations of motion [Eq. (12)], and the thick solid line indicates it. The analytical results match the numerically computed growth rates derived by the full solution of the system of linearized equations.

For clarity, the general mechanism of tachyonic instabilities and their distinction from the single-field case with unstable modes at $k = 0$ is reviewed in Appendix A.1, which explains how the finite-momentum instability bands seen in Fig. 4 arise in the two-component condensate system.

### 3.2.1 Physical interpretation

We now provide a physical interpretation of the instability. Because the first condensate has a higher energy than the second, the tunneling of quasiparticles into the lower-energy well releases excess potential energy, which is converted into kinetic energy. Since the initial momentum is approximately zero, energy conservation requires the creation of a pair of quasiparticles with opposite momenta that share this released energy (similar to twin-atom beams [49]). This process gives rise to a well-defined instability band in the momentum spectrum. The two resulting momentum peaks can be understood as signatures of pair emission: one or both quasiparticles tunnel into the lower-energy condensate, while momentum conservation enforces their back-to-back emission with total momentum zero. The resulting correlated pair can be experimentally detected via opposite-momentum correlations (see Sec. 5.5). There are exactly two main processes allowed by energy conservation:

1. *Single tunneling*: one quasiparticle remains in the original well while its partner tunnels to the second well.

2. *Pair tunneling*: both quasiparticles tunnel together into the lower-energy condensate.

We will refer to the processes as $(1,1) \to (l,m)$ for two particles starting from the first condensate and produced in the wells $l$ and $m$, where $l, m = 1, 2$. Note that in contrast to the

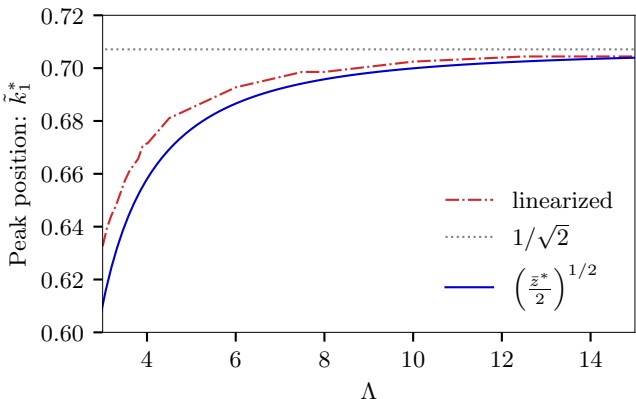

Figure 5: Prediction of the primary peak location at momentum $\tilde{k}_1^*$ versus the dimensionless ratio $\Lambda$. The analytical [Eq. (22)], numerical, and asymptotic results are indicated by a solid, dashed-dotted, and a dotted line, respectively.

twin beam experiment [50], no further selection rules apply here, such that all processes that conserve energy are allowed. Furthermore, note that in zero spatial dimensions, this is not possible due to energy *and* momentum conservation.

A simple perturbative calculation treating the tunneling term as a perturbation can help gain further physical insight. First, let us examine more in detail the $(1,1) \rightarrow (1,2)$ process, shown in Fig. 6a). The dominant contribution to particle creation comes from the momentum mode satisfying the resonant condition

$$\epsilon_1(\tilde{k}) + \epsilon_2(\tilde{k}) = \frac{\Delta\mu}{\mu}, \tag{20}$$

which equals the sum of energies of the particles each excited in one well, given by the Bogoliubov dispersion relation for the case of a free condensate

$$\epsilon_j(\tilde{k}) = \sqrt{\tilde{k}^2 \left( \frac{\mu_j}{\mu} + \frac{\tilde{k}^2}{4} \right)}, \tag{21}$$

to the scaled difference of chemical potentials $\Delta\mu/\mu = (\mu_1 - \mu_2)/(\mu_1 + \mu_2)$. The solution of Eq. (20) gives the resonant momentum

$$\tilde{k}_1^* = \left( \frac{\bar{z}_0^*}{2} \right)^{1/2} = \left( \frac{1}{4} - \frac{1}{\Lambda} \right)^{1/4}, \tag{22}$$

that corresponds to the center of the first instability peak in the bottom plot in Fig. 4. Therefore, the specific location of the peak only depends on the specific choice of the dimensionless parameter $\Lambda$. A comparison between the analytical and numerical results can be seen in Fig. 5. The analytical prediction [Eq. (22)], shown with dashed lines and detailed in Ref. [36], is compared to the numerical result obtained by solving the linearized equation and fitting the growth rate, shown with a solid line. The deviation from the analytical, theoretical prediction is due to numerical uncertainty and the approximation to a weakly tunnel-coupled system. Asymptotically, for $\Lambda \gg 1$, $\tilde{k}_1^* \rightarrow 1/\sqrt{2}$, indicated as a dotted horizontal line in the figure.

The second process which is energetically allowed is $(1,1) \rightarrow (2,2)$, shown in Fig. 6b). Energy conservation gives the resonant condition

$$2\epsilon_2(\tilde{k}) = 2\frac{\Delta\mu}{\mu}. \tag{23}$$

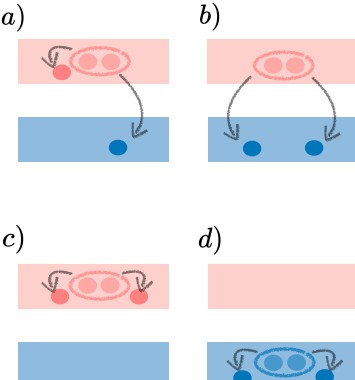

Figure 6: Physical processes related to the primary instability of the coupled BEC system with components $\Psi_1$ and $\Psi_2$, indicated by the upper and lower elongated shapes, respectively: $a$) excitation of a pair of particles from the first condensate, where one particle further tunnels to the second condensate; $b$): excitation of a pair from the first condensate that tunnels to the second one; $c$) and $d$): excitation of a pair of particles from each of the two condensates.

Proceeding in a similar fashion as in the previous case, the solution of Eq. (20) gives the resonant momentum $\tilde{k}_2^*$ that corresponds to the center of the second instability peak in Fig. 4.

In both cases, we obtain the same results for the peak location for the instability as discussed in Ref. [25] for the limiting case of large $\Lambda$, even though the reference studies the different dynamical regime of MQST. In fact, the actual physical process of pair creation and redistribution in the wells is identical to our case.

### 3.3 Linearization for oscillating mean fields: parametric instabilities

After discussing the linearization around the stationary point in $(\phi, z)$ space, we now want to generalize this approach to generic (time-dependent) closed trajectories around it. In this case, there is an additional instability, parametric resonance, caused by the oscillatory behavior of the mean fields. The oscillations around the minimum of the effective potential act as a source for a parametric resonance instability, in addition to the tachyonic instability discussed before. In fact, an entire range of modes experiences a positive growth rate, and the width of this instability band increases with the modulation amplitude $\Delta z = z_0 - z_0^*$. First, we get a qualitative understanding of the mechanisms behind the additional peaks.

#### 3.3.1 Physical interpretation

Once again, we can use energy arguments, as explained below, to gain an intuitive understanding of the underlying physical processes that lead to the different peaks. We will treat the condensate as very weakly coupled to use the free dispersion relation. This procedure is justified because the ratio $1/\Lambda$ is small. In the following, we explain the origins from resonance conditions. As in the previous subsection, we will indicate the process where two particles from condensate $j$ are excited, and end up in the $l$ and $m$ wells as $(j, j) \xrightarrow{r} (l, m)$, where the 'r' is indicating that the mechanism behind the excitation of the pair is parametric resonance. The allowed processes are the following:

- $(j, j) \xrightarrow{r} (j, j)$

  These two cases correspond to a pair of particles leaving a condensate in a given well $j = 1, 2$, which gets excited by higher momentum modes in the same well. The resonance

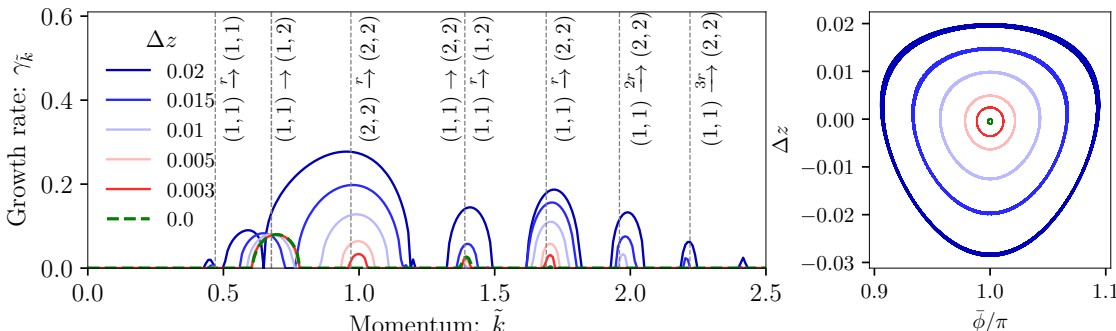

Figure 7: Instability chart as a function of the wavenumber $\tilde{k}$ for the *oscillating* $\pi$-trapped state with varying initial conditions $\Delta z$. The oscillations and amplitude of the mean fields are shown in the inset and legend, respectively. We choose $\Lambda = 7.10$ for better comparison with the analytical predictions for the location of the peaks, see the main text for details. The vertical dashed lines indicate the position of the maximum of the instability band from the analytical calculation. The tachyonic instabilities $(1,1) \rightarrow (1,2)$ and $(1,1) \rightarrow (2,2)$ are indicated with dashed lines. Parametric instabilities from left to right: leading resonance $(1,1) \xrightarrow{r} (1,1)$ and leading resonance $(2,2) \xrightarrow{r} (2,2)$ [Eq. (24)], resonance $(1,1) \xrightarrow{r} (1,2)$ [Eq. (28)], leading resonance $(1,1) \xrightarrow{r} (2,2)$ [Eq. (27)], second order resonance $(1,1) \xrightarrow{2r} (2,2)$, third order resonance $(1,1) \xrightarrow{3r} (2,2)$.

occurs when the Bogoliubov dispersion relations [Eq. (21)] match the frequency $\omega_r$ of the oscillation frequency of the imbalance densities $\bar{\rho}_j$ (which equals the oscillation frequency of the trapped relative phase $\bar{\phi}$ since they are conjugate variables). In this case, energy conservation yields

$$2\epsilon_j = \omega_r. \tag{24}$$

The same result for the location of the unstable modes can also be estimated in a different way, which connects with the usual discussion of parametric resonance in field theory. Specifically, for small oscillation amplitudes (i.e., when $\Delta z \ll 1$), by taking a further time derivative of the density equations in Eq. (16) and taking the limit of uncoupled condensates, and slowly varying mean fields, the resulting equations can be brought to the form of a Mathieu equation [51, 52] in terms of the $\delta\rho_j$ alone,

$$\left[ \frac{\partial^2}{\partial s^2} + A_{\tilde{k}} - 2q_{\tilde{k}} \cos(2s) \right] \delta\rho_j(s, \tilde{k}) = 0, \tag{25}$$

with the dimensionless time $s = \omega_r \tau/2$, and parameters $A_{\tilde{k}} = \epsilon_j^2(\tilde{k})/(\omega_r/2)^2$ and $q_{\tilde{k}} = \Delta z \epsilon_{\tilde{k},0} \mu_j/(\omega_r/2)^2$. Equation (25) admits oscillatory solutions with exponentially growing amplitudes that describe parametric resonance. The width of this instability band is delimited by the modes satisfying

$$A_{\tilde{k}} = 1 \pm q_{\tilde{k}}, \tag{26}$$

which increases with the amplitude $r$ of the oscillation. That is, resonance occurs for those momentum modes whose energy equals half a quantum of energy $\hbar\omega_r/2$ injected in the system through the oscillation at frequency $\omega_r$. Note that the same type of Mathieu equation is discussed in Ref. [23] to describe the higher-order resonances. We note that the Mathieu equation governing the instability coincides with our Eq. (25). However, while in Ref. [23] the instability arises from an external drive implemented through

explicit modulation of the tunneling coupling (thereby enforcing parametric growth), here it is the oscillations of the mean field that act as the periodic drive.

For clarity, the general mechanism of parametric instabilities and their relation to Mathieu-type resonance equations is reviewed in Appendix A.2, which explains how the oscillating mean fields in our system give rise to the finite-momentum resonance bands identified here.

- $(1, 1) \xrightarrow{r} (2, 2)$
  It can also occur that from the first condensate, a pair of atoms gets excited to the second condensate. In Fig. 6, $c$) and $d$) represent the physical process of two particles leaving the condensate end and getting excited at opposite momenta from

$$2(\epsilon_2 - \frac{\Delta\mu}{\mu}) = \omega_r. \tag{27}$$

- $(1, 1) \xrightarrow{r} (1, 2)$
  The last case occurs when a pair of particles from the first condensate gets excited by the driving force of the oscillations, and only one atom tunnels to the other condensate. In this case,

$$\epsilon_1 + \epsilon_2 = \omega_r + \frac{\Delta\mu}{\mu}. \tag{28}$$

The resonance peak overlaps with the tachyonic peak discussed in the previous subsection. Still, we see that for increasing oscillation amplitude, the corresponding growth rate first appears even without oscillations being present ($\Delta z = 0$), then diminishes, and then takes over growing in amplitude.

Moreover, for each resonance process, the above analysis only concerned the first (order) instability band. Still, in general, multiples of the resonance, so higher frequency harmonics are also in resonance. Each band in momentum space has a width of order $\delta\tilde{k} = l^2$, $l \in \mathbb{N}$. So they will be located at higher frequencies (but not multiples), with a narrower width, and a lower growth rate. We denote these processes in the figure as $(i, j) \xrightarrow{lr} (i, j)$ to indicate the $l$-order resonance.

In general, for increasing the oscillation amplitude $\Delta z$, the width of the parametric instability bands increases and the bands become higher. There are no further resonance processes apart from the ones discussed. For instance, the process involving a pair of particles from the second condensate with one (or more) particles tunneling to the second condensate is energetically forbidden. The instability chart is summarized in the left panel of Fig. 7, along with the evolution of the mean fields displayed in the right panel. Energetically possible processes are represented by vertical lines. The dashed green line illustrates the limiting case where only tachyonic instabilities are present, occurring when the amplitude of the mean fields' oscillation approaches zero.

## 4   Beyond linearization with numerical simulations

The previous section gave a linearized analysis of the evolution of perturbations, allowing us to gain an analytic understanding of the primary instabilities. In this section, we solve the full GPE equations with initial fluctuations as a seed for the instabilities. We refer to Appendix C for a review of the method and the implementation details. In order to observe the instabilities more clearly, we first reduce the noise to $\eta = 0.001$ (see Eq. (C.1) for the definition of the noise factor $\eta$). Later in Sec. 4.1, we will examine the case of vacuum and thermal noise.

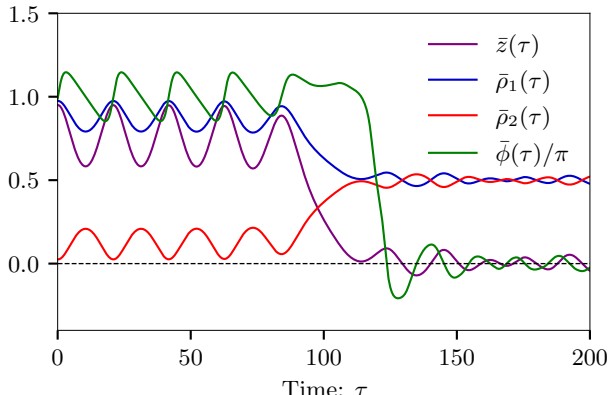

Figure 8: Time evolution of the densities in the two condensates $\bar{\rho}_j(\tau)$, relative imbalance $\bar{z}(\tau)$ and relative phase $\bar{\phi}(\tau)$. The $\pi$-trapped state was initialized in $(z_0, \phi_0) = (0.95, \pi)$ with $\Lambda = 3.55$. The oscillations of the fields are followed by the equilibration of the imbalance and relative phase to zero at late times (dashed line).

For all simulations, we have considered typical values for an experiment using $^{87}$Rb: $m = 87$ amu, $a_s \approx 5.2$ nm, the longitudinal trapping potential $\omega_\perp = 2\pi \nu_\perp$, $\nu_\perp = 1.4$ kHz. We consider a homogeneous system ($\omega_\parallel = 0$). The tunneling coupling $J = 2\pi \nu_J$, has been set to $\nu_J = 200$ Hz. Furthermore, we consider 20000 particles and 2048 lattice points with equal spacing $a = 0.1~\mu$m, corresponding to typical experimental densities $\rho \approx 100$ atoms per $\mu$m. In our classical-statistical simulations, we typically average over at least 200 runs.

As in the previous section, we first present results for initial conditions for the fields at the saddle point and then generalize to small oscillations around it. These numerical simulations enable us to study also the non-linear dynamics: In Fig. 8, the time evolution of the fractional imbalance $z$ and relative phase $\phi$ is shown. At late times, the system deviates from the classical trapped trajectory, showing damping [8] and eventually approaching equilibrium where both the relative phase and the imbalance vanish. As $z$ and $\phi$ are canonically conjugate variables, density fluctuations reach their maximum when phase fluctuations cross zero and vice versa.

The relevant observable for the study of the growth of fluctuations is the momentum distribution

$$\rho_j(\tau, \tilde{k}) = \frac{1}{L} \overline{\Psi_j(\tau, \tilde{k}) \Psi_j^*(\tau, \tilde{k})}, \tag{29}$$

where $\Psi_j(\tau, \tilde{k}) = \int \mathrm{d}\tilde{x} \Psi_j(\tau, \tilde{x}) e^{-i\tilde{k}\tilde{x}}$ denotes the Fourier transform of the field and the notation $\overline{(\dots)}$ refers to the ensemble average across all realizations. The momentum distribution is shown in Fig. 9 for the two cases with identical parameters as examined earlier.

The simulations show the primary instabilities at early times already discussed using the Bogoliubov approximation in Sec. 3. As the occupation number of unstable modes increases exponentially, they become highly occupied, and the system enters a non-linear regime. The Bogoliubov approximation breaks down, and the stage of secondary amplification sets in. As time progresses, the system deviates from the linear regime, which we discuss next. Our analysis follows a similar approach as in Ref. [39].

The emergence of both primary as well as secondary instabilities can be observed from the results of the numerical lattice simulations shown in Fig. 9. The black dashed lines show the predicted peaks according to energy conservation (as already discussed in Sec. 3.3.1). The peaks correspond to the maximally unstable modes of the linearized theory within each instability band (cf. Fig. 7).

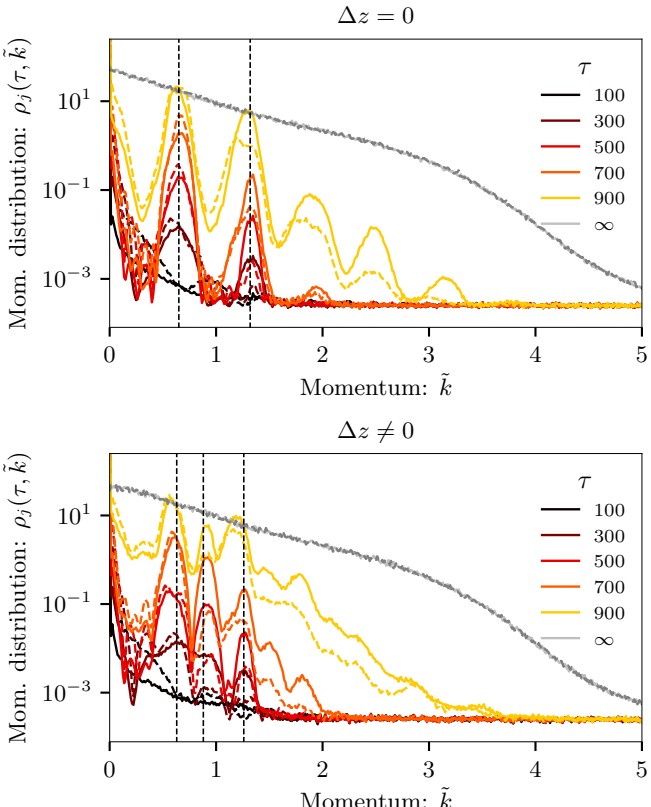

Figure 9: Momentum distribution of the condensates (solid and dashed lines correspond to the first and second condensate, respectively) for the $\pi$-trapped state at different times. The black dashed lines show the maximally unstable momentum modes obtained using energy considerations (see Sec. 3.2.1). *Upper:* Fields initialized in the mean field stationary point $(\phi_0, z_0) \approx (\pi, z_0^*)$. *Lower:* Fields initialized in $(\phi_0, z_0) = (\pi, 0.95)$, corresponding to weak oscillations around the mean-field stationary points, i.e., $\Delta z \neq 0$. These initial conditions cause the same instabilities as the $\Delta z = 0$ case, but additional peaks arise due to oscillations in phase and imbalance that cause parametric resonance.

Figure 10 shows the time evolution of the most unstable modes corresponding to the primary and secondary peaks in the first wire. The dotted blue and dashed-dotted red curves represent the primary unstable modes, while the black dotted and green solid curves correspond to the secondary unstable modes with enhanced growth rates. As it can be observed, the secondary modes begin to grow later, around $\tau \approx 600-800$, but then grow at a faster rate than the primaries. The straight lines represent exponential growth with a corresponding growth rate $\gamma_{j,lin}$, where $j = 1, 2$ indicates the prediction by the linearized theory, corresponding to momenta $\tilde{k}_j^*$. It's worth noting that the result from the linearized theory is slightly higher than the GPE, but the presence of noise can account for this difference.

When macroscopically occupied modes start interacting, the mode with momentum $3\tilde{k}^*$ starts growing due to the dominant scattering process $(\tilde{k}^*, \tilde{k}^*) \to (-\tilde{k}^*, 3\tilde{k}^*)$ [39]. Furthermore, the momentum modes at $\pm\tilde{k}^*$ can also engage in interactions with the condensate mode of $\Psi_1$. This process is expected to increase the occupation number of the $2\tilde{k}^*$ mode [39].

Over time, the process of the secondary instabilities continues, resulting in the occupation of progressively higher modes. An alternative approach to study these secondary instabilities is, rather than letting the time evolution fill these excited modes, to directly seed the primary

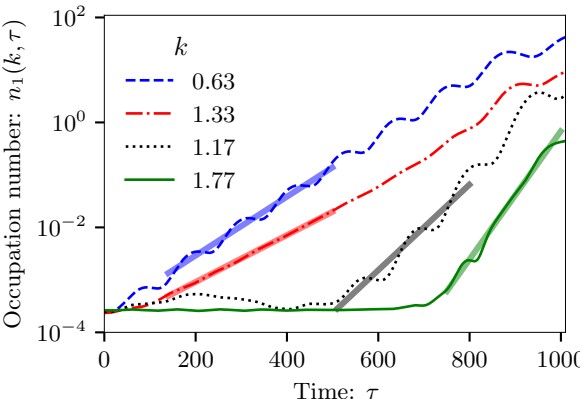

Figure 10: Time evolution of the occupation number of primary and secondary exponentially unstable modes in the first wire. The dashed and dotted-solid lines correspond to the two maximally unstable modes, also shown in Fig. 9. The straight lines indicate the corresponding growth rate $\gamma_{j,lin}$ as predicted by the linearized theory. The black-dotted and green solid curves are the secondary unstable modes with enhanced growth rates.

modes, as discussed in Ref. [39]. At later times, the growth of all modes deviates from the exponential growth and then eventually stops, as no single process dominates the dynamics. In this final phase, the distinct peak structure of the spectrum is lost, particles spread across various momentum modes, and the system undergoes thermalization (represented by gray lines for $\tau \to \infty$ in Fig. 9).

## 4.1  Finite Temperature

One of the major experimental limitations for observing the presented dynamics is the amount of noise in the system. At the beginning of this section, we considered a reduced noise level to make the different stages of the evolution visible. For increasing noise, this clear separation into periods of linear primaries, a limited number of non-linear secondaries, and a late stage of complete non-linear thermalization disappears. This, in general, presents one of the major challenges in experimentally observing secondaries in analog field theory simulations, with a recent measurement for classical water-air interfacial waves [53].

We display in Fig. 11 the momentum distribution of the empty well $\rho_2$ at $t = 3.8\,\text{ms}$ for different initial temperatures $T = 0 - 50\,\text{nK}$. The empty well was initialized by sampling the quantum noise within the quasiparticle basis (see Appendix C for more details on the numerical implementation). Around this timescale, the primary instability peaks are clearly visible at $T = 0$, with the two momentum peaks located at $\tilde{k}_1^*$ and $\tilde{k}_2^*$. The primary instability peak located at $\tilde{k}_1^*$ remains visible and in good agreement with theoretical predictions for temperatures easily achievable with current cold-atom experiments. Note that for even higher temperatures, we find a slight shift of the instability peaks to higher momenta.

Detecting the relevant higher-loop processes experimentally could be difficult at higher temperatures. To overcome this limitation, one approach is to seed the primary instabilities, as done, e.g., in Ref. [39], and/or further decrease the temperature [54, 55]. Further, the decay processes are expected to create excitations in pairs, leading to higher-order correlations between different momentum modes, similar to the studies performed in Refs. [50, 56]. A detailed analysis of the pair-correlations and their decoherence due to finite temperature is beyond the scope of this paper.

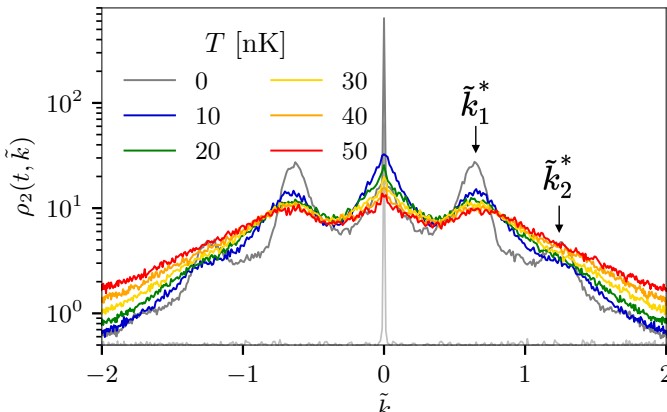

Figure 11: Momentum distribution in the second well $\rho_2$ for temperature $T = \{0, 10, 20, 30, 40, 50\}$ nK at $t = 3.8$ ms (this timescale corresponds in dimensionless units to $\tau \approx 300$ in Fig. 9).

## 5 Experimental feasibility

In this section, we outline the requirements needed to fulfill to investigate experimentally the $\pi$-state dynamics studied in this work. The parameter regimes required to be reached can be inferred from Figs. 2 and 3, and Eqs. (9) and (10). These define the necessary initial conditions $z_0$ and $\Lambda \in (\Lambda_b(z_0), \Lambda_u(z_0))$ for the appearance of the $\pi$-oscillations in the dimensionless $(\Lambda, z_0)$ spacefor $\langle z \rangle_t > 0$. These specific parameter regimes are readily implementable in present-day experiments with quasi-1D Bose gases on AtomChips [36].

As discussed above (see Sec. 3), the stability of the $\pi$-state dynamics is determined by the appearance of non-trivial scattering solutions related to the emission of correlated pairs of excitations. Increasing the system size $L$ along the extended direction of the bosonic Josephson junction, the system transitions from stable $\pi$-oscillations, as predicted by the two-mode (mean-field) model, to the reported instabilities.

### 5.1 Validity of the 1D model under experimental conditions

Our analysis so far has been based on an idealized, strictly 1D scenario. The one-dimensional approximation we considered neglects interaction-induced modifications (broadening) of the ground state wavefunction in the radial directions [57,58], the system becomes non-integrable and described by the non-polynomial Schrödinger equation [57]. Consequently, a more realistic one-dimensional model leads to a non-trivial density dependencies of the coupling constants (see e.g. [37]). The resulting dominant effect on the dynamics of the linearized fluctuations is a small renormalization of the tunnel-coupling $J$ and the speed of sound $c_s$. While the former only needs to be considered when determining the correct parameter regime (i.e., the dimensionless $\Lambda$), the latter simply leads to a trivial shift of the resonances.

The validity of the one-dimensional model Eq. (1) [58–60], as well as its low-energy effective description in terms of the sine-Gordon model [20], has been shown in previous Atomchip experiments. In particular, this has been shown for Josephson oscillations [30, 31], Floquet engineered bosonic Josephson junctions [61], and the effective low-energy description for balanced condensates [29, 62, 63].

Another imperfection in relation to our theoretical models is the longitudinal confinement potential and the finite size of the experimental system. Even though Digital Micromirror Devices (DMDs) enable precise control over the shape of the longitudinal trapping potential

on the AtomChip [64–66] allows to implement homogeneous box traps and the dynamics within the bulk is well approximated by a homogeneous system, the finite size of the trap is felt at timescales larger then propagation speed of fluctuations originating from the boundary.

## 5.2   State preparation

A detailed analysis of the dynamics involved in preparing the initial state goes beyond the scope of this paper nevertheless, we will argue that a combination of already established and often applied techniques will allow for experimental implementation of the $\pi$-state. Atomchips [67–69] enable precise creation and control of a double-well (DW) potential for 1d quantum gases [70–72] on timescales much faster than the typical timescale of the reported instabilities. In particular, this includes: (i) variation of the tunneling coupling $J$ (see e.g. [29]), (ii) tilting or rotation of the DW potential [70], and (iii) Floquet engineering [61].

The precise control of the DW potential facilitates a number of prospective preparation sequences. Reaching the desired dimensionless parameters $(\Lambda, z_0)$ is a matter of designing the loading sequence of the DW. Tilting allows to set the initial imbalance $z_0$ when loading the atomic cloud in the DW. With the total atom number and the initial imbalance $z_0$ known, adjusting the tunneling $J$ will then allow for dialing in the desired $\Lambda$. This way, we have independent control of the initial state in the $(\Lambda, z_0)$ space. Applying optimal control techniques, similar to the ones recently demonstrated for regular splitting into a 1D DW [73], will allow us to establish fast and reliable preparation of the initial state with minimal disturbance.

Finally, the preparation of the initial relative phase difference will allow for the preparation of the $\pi$-state, as opposed to MQST. Coherent splitting of a single 1D quasi-condensate into a DW sets the initial relative phase between the two condensates close to zero. An additional subsequent tilt of the DW potential can be used to tune the desired phase difference between the split quasi condensates [30, 36], and thereby populating the $\pi$-state. This phase imprinting by tilting the DW potential can be directly integrated in the optimal control sequence for preparing the $(\Lambda, z_0)$ state. A different approach reaching the $\pi$-state, relies on imprinting the initial relative phase using a DMD.

Leaves the thermal fluctuations in the quantum gas as a final consideration for the preparation of the $\pi$-state. The initial temperature of the quantum gas will determine the global fluctuations of the system. The $\pi$-state itself will see the fluctuations between the two tunnel coupled quantum gases. These can be dramatically reduced using specific splitting ramps [31, 74] or by optimal control techniques as proposed in [75, 76].

A completely different pathway is through Floquet engineering the bosonic Josephson junction [61], which allows to adjust the initial relative phase difference between the two condensates. Varying its magnitude changes the Floquet assisted effective tunneling rate, and hence the relevant parameter $\Lambda$. While in theory this method to prepare the $\pi$-state is comparatively simple, in experiment, Floquet driving might increase the amount of noise present in the system.

In addition to mean-field observables, fluctuations provide crucial information: enhanced variance in the imbalance or deviations from sinusoidal phase dynamics are expected hallmarks of the onset of instabilities. Quantifying such higher moments will therefore allow one to discriminate between regular Josephson oscillations and instability-driven dynamics.

## 5.3   Relevant observables and their measurement

Cold atom quantum gas experiments have the advantage that there are many techniques to probe the quantum state of the system under study available. If one experiments with single systems, then the experiment delivers not only the expectation value of an observable but also its full distribution function (= full counting statistic) [77], and higher order correlation

functions [29], which both contain detailed information about the system. In an elongated 1D system, all these observables can be obtained as a function of the coordinate along the length of the system.

The simplest and most straight forward observables to study the $\pi$-state are the time evolution of the (local) imbalance $z(t) = n_1 - n_2$ and (local) phase $\phi(t)$. The ensemble average of many experiments then corresponds to the (local) mean-fields $\bar{z}(t)$ and $\bar{\phi}(t)$, as presented in Fig. 8. In addition the experiments delivers in addition the full counting statistics of these averaged observables.

A second insightful observable for studying the specifics of the dynamics and decay of the $\pi$-state is the momentum distribution of excitations, which becomes visible after a long time of flight (see twin atom experiment [49, 50] and Fig. 13) or through condensate focusing [78].

In cold-atom quantum gas experiments, the time evolution is commonly measured via repeated destructive measurement of observables. A measurement cycle consists of:

(i) Preparation of the initial state: Initializing the $\pi$-state.

(ii) Holding the quantum gas in the trap for a predetermined holding time $t_{hold}$ during which the system evolves with (unitary) dynamics.

(iii) Releasing the atoms by switching off all external trapping potentials, and applying an optional additional manipulation to select the observable to be measured.

(iv) Letting the system evolve freely for an additional time-of-flight $t_{\text{ToF}}$, which can be short for 'in situ' observables, is medium when looking at interference and can be very long when measuring momentum.

(v) Finally the (local) atomic density is measured by taking an image. The desired observable is then extracted from the image of the atomic density.

This cycle is then repeated many times, and the expectation values (and the full distribution function) of the observable are obtained from many independent realizations. The time evolution is then reconstructed by varying the holding time $t_{hold}$.

If in step (iii), the two clouds are simply released from the trap, they expand and overlap transversely in time of flight, and a measurement of atomic density gives direct access to the spatially-resolved relative phase between the two condensates through matter-wave interference [79–81]. The spatially resolved interference along the extended direction allows the extraction of the occupation of the fluctuations in the quantum gas. The higher-order correlations give insight into the interactions of the excitations [29, 62].

From these measured interference pattern the spatially resolved fluctuations in the common mode can be extracted following [82]. The spatially resolved relative density fluctuations are, in principle, accessible through full tomography of the relative degree of freedom [83].

If in step (iii), the atoms are given an additional outward (in the double-well direction) momentum kick, the two clouds separate, and one can measure the spatially-resolved density in the left and right well. This allows to extract the spatially-resolved density imbalance directly. Employing single atom sensitive fluorescence imaging [84], these measurements can have an atom noise far below shot noise and reveal number squeezing and entanglement between the two clouds [31, 73, 74].

Recent developments combining these two techniques through partial out-coupling of atoms, which allows simultaneous measurement of both quadratures [85], give direct access to the Husimi-distribution of the non-commuting relative density and phase variables.

The individual momentum distributions of the atoms in the left and right well can be measured after a long time-of-flight $t_{\text{ToF}} \approx 46\,\text{ms}$ in a light-sheet with single-atom sensitivity [49, 50, 84]. Note that due to the rapid expansion in the tightly confined radial directions,

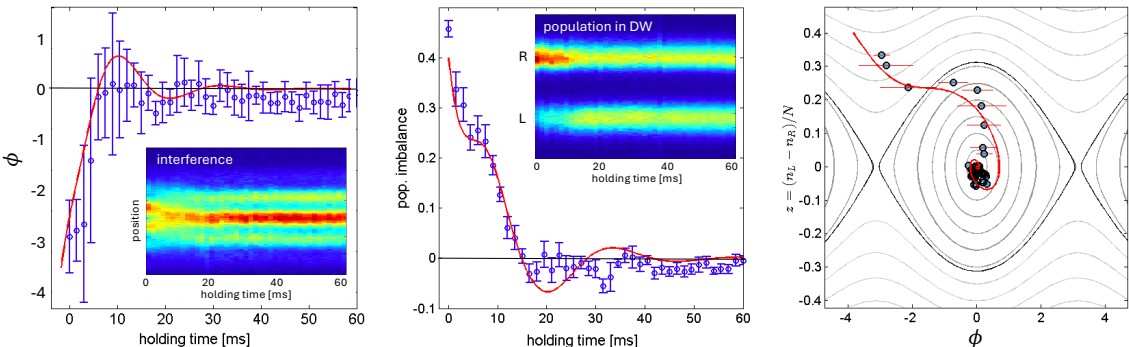

Figure 12: Decay of macroscopic quantum self trapping (MQST) to a phase locked state with $z \approx 0$ and $\Phi \approx 0$. *Left*: Relaxation of the phase $\Phi$. *Center*: Relaxation of imbalance measured by applying a transverse kick. *Right*: Trajectory of the decay in the $z_0$-$\Phi$ plane. *Inserts*: Time evolution of the interference pattern (atomic patterns representing the left and the right wells) put together from the original images. Data from experiments carried out during the thesis of M. Pigneur [36].

the interactions switch off very fast and for $t_{\text{ToF}} \gtrsim 1$ ms the time-of-flight evolution can be well approximated by freely propagating atoms and hence conserves the momentum distribution along the extended direction. Additionally, the effective momentum resolution of the experiment can be further enhanced into the infrared through condensate focusing [78], which results in a perfect mapping of momentum on position in the imaging plane.

Finally, beyond single-particle spectra, a decisive experimental signature is the appearance of correlations between atoms with opposite longitudinal momenta. Such back-to-back correlations, directly analogous to the twin-atom beams observed in Ref. [49, 50], would constitute a smoking-gun signal of pair emission from the unstable condensate. Measuring second-order correlation functions $g^{(2)}(k, -k)$ in the time-of-flight images would thus provide unambiguous evidence for the predicted tachyonic and parametric instabilities.

## 5.4 Illustration of the accessible observables

We now highlight the experimental feasibility of observing the details of the physics involving the dynamics and decay of the $\pi$-state. To illustrate the different observables one can employ, we show here data from initial exploratory experiments probing a very similar physical situation: the decay of macroscopic quantum self trapping (MQST).

Figure 12 shows a typical experimental run probing the dynamics and decay of MQST. From the atom images we extract in one experimental run the interference pattern and in a separate experimental run, employing the transverse kick, the atomic density. From there, we build the interference and population 'carpets' (inserts in Fig. 12). From the extracted phase and imbalance, we can reconstruct the decay path as plotted in the right-most graph. A similar measurement protocol can be applied to study the dynamics and decay of the $\pi$-state.

Figure 13 shows a test experiment where we measured the longitudinal momentum distribution after a long (46 ms) time of flight and compare a regular Josephson oscillation (left) with the decay of MQST (right). The measured momentum distribution in the case of the decay of MQST shows clearly two opposite shoulders, indicating the pairs that carry away the potential difference between the two wells in their kinetic energy. The data is not good enough to show sub-shot noise correlations, which would be a clear signature of the pairs. Employing condensate focusing [78] will make the central peak much narrower and the momentum peaks associated with the excitations produced in the decay of MQST more pronounced and background free. A similar measurement protocol can be applied to study the momentum

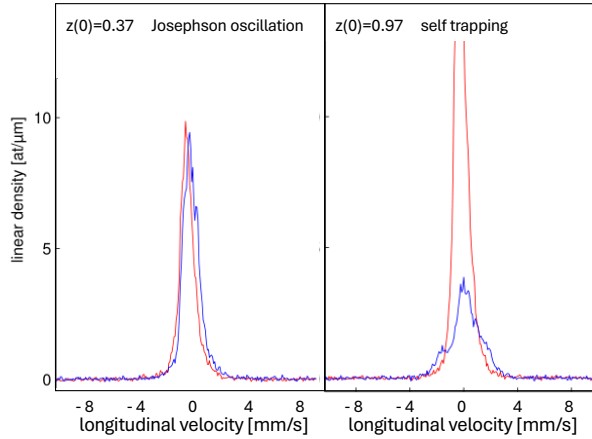

Figure 13: Longitudinal momentum distribution. *Left*: Josephson oscillation regime (red: initial; blue: after a full oscillation). *Right*: MQST regime. Halfway through the decay of the imbalance (blue), two shoulders left and right from the central peak indicate the two opposite momenta of the pairs of excitations, which carry away the potential difference between the two wells in their kinetic energy. Data from experiments carried out during the thesis of T. Berrada [35].

distribution of the excitations created in the decay of the $\pi$-state.

## 5.5   Correlation measurements as key signatures

While mean-field observables such as the imbalance $z(t)$ and the relative phase $\phi(t)$ already provide valuable insight into the decay of the $\pi$-state, the most decisive signatures of the underlying instabilities are encoded in fluctuations and correlations. Enhanced variance in the imbalance or deviations from sinusoidal phase oscillations are expected hallmarks of the onset of instability-driven dynamics. Quantifying such higher moments allows one to discriminate between regular Josephson oscillations and dynamics triggered by tachyonic or parametric instabilities.

A particularly striking observable is the momentum-resolved two-particle correlation function

$$g^{(2)}(k, -k) = \frac{\langle \hat{n}_k \hat{n}_{-k} \rangle}{\langle \hat{n}_k \rangle \langle \hat{n}_{-k} \rangle}, \tag{30}$$

which directly probes correlations between atoms with opposite momenta. The unstable quasiparticle pairs produced during the decay of the $\pi$-state are emitted back-to-back, such that $g^{(2)}(k, -k)$ is strongly enhanced in the unstable momentum bands. This situation is directly analogous to the twin-atom beams observed in Refs. [49, 50], where correlated pairs emerged from a driven instability in an elongated condensate. In the present context, the amplification of correlations between $\pm k$ modes constitutes a smoking-gun signature of both tachyonic and parametric instabilities. While primary instabilities are most accessible, secondary processes leading to higher-momentum excitations (Sec. 3) could in principle be detected via extended correlation analyses, albeit with more stringent requirements on noise and statistics.

State-of-the-art fluorescence imaging with single-atom sensitivity and large statistics allows the extraction of both momentum distributions and correlation functions from the same dataset. In particular, detecting sub-shot-noise correlations between $\pm k$ modes would establish a direct parallel to non-classical pair production processes in quantum optics and cosmology. In this way, the cold-atom realization of extended bosonic Josephson junctions provides

an experimentally accessible platform to observe universal instability mechanisms and their associated correlation signatures in real time.

## 6 Conclusions

In this work, we investigated the dynamics of two weakly tunnel-coupled quasi-one-dimensional Bose gases in the $\pi$-state. While the system exhibits $\pi$-oscillations classically under the condition that the ratio of the chemical potential to the tunneling energy is sufficiently large and the initial energy is above a critical value, this state becomes unstable due to quantum fluctuations.

We studied the early time dynamics by linearizing the theory and observed that the state deviates from the mean-field prediction. At first, we examined the simpler case of the system initialized such that the oscillations are suppressed, and the fields remain constant, identifying tachyonic instabilities. Secondly, we also generalized our analysis to oscillating fields, producing additional parametric resonance instabilities. Physically, the instability is due to the excitation of quasiparticle pairs from the two condensates to characteristic bands of momenta. The occupation of these modes grows exponentially, and the momentum peaks can be predicted analytically using energy considerations. We analyzed in detail the different mechanisms underlying the pair production.

Later in the dynamics, the system develops a very sharp momentum distribution, where the linear theory breaks down. We go beyond linearization by means of GPE numerical simulations, in particular, to study the formation of secondary instabilities. Eventually, at late times, the system generally reaches a steady state, at which point the momentum distributions in the two wires become the same, and the relative phase vanishes.

Our analysis provides detailed results on the instability dynamics and decay of the $\pi$-trapped state. This provides an important prerequisite for possible experimental realizations in a BEC system. To this end, our analysis of finite-temperature initial states shows that the primary instability peaks remain robust within experimentally achievable regimes, underscoring the immediate feasibility of observing these effects.

Looking ahead, our analysis establishes a framework for probing fundamental instability mechanisms in a controlled cold-atom setting. The $\pi$-state dynamics of an extended bosonic Josephson junction provide a unique opportunity to connect condensed-matter realizations of tachyonic and parametric instabilities to paradigms familiar from high-energy physics and cosmology, such as spinodal decomposition and preheating after inflation. Beyond these fundamental links, the ability to prepare, control, and detect correlated quasiparticle pairs in real time paves the way toward quantum simulation of early-universe scenarios and nonequilibrium field theory phenomena in the laboratory. In particular, correlation measurements between opposite-momentum modes, as outlined in Sec. 4, provide a smoking-gun experimental signature directly linking our predictions to feasible detection schemes. Experimentally, sub-shot-noise correlations in the emitted twin beams, accessible with state-of-the-art imaging techniques, would offer a striking signature of these instabilities. More broadly, exploring the crossover from linear instabilities to nonlinear thermalization in such systems may open new avenues for designing quantum devices, engineering entangled matter-wave sources, and benchmarking theories of far-from-equilibrium quantum dynamics.

## Acknowledgements

We thank Stefan Floerchinger, Michael Heinrich, Eduardo Grossi, Louis Jussios, Silke Weinfurtner, and Tiantian Zhang for valuable discussions.

**Funding information**    This research was supported by the DFG / FWF under the Collaborative Research Center SFB 1225 ISOQUANT (Project-ID 27381115, Austrian Science Fund (FWF) I 4863) and the the European Research Council: ERC-AdG "Emergence in Quantum Physics" (EmQ) under Grant Agreement No. 101097858. S.E. acknowledges additional support by the Austrian Science Fund (FWF) [Grant No. I6276, QuFT-Lab].

# A  Dynamical instabilities

In quantum field theory out of equilibrium, instabilities are characterized by exponential amplification of quantum fluctuations. These behaviors typically manifest in two-point correlation functions. This mechanism leads to rapid amplification of low-energy fluctuations. Exponentially growing correlations can arise via different physical mechanisms. In the following, we introduce the two different types that arise in our model presented in Sec. 3.

## A.1  Tachyonic instability

The *tachyonic instability* is often referred to as spinodal decomposition, or simply spinodal instability. Consider a single-component real scalar field $\varphi$ with potential

$$V(\varphi) = -\frac{m^2}{2}\,\varphi^2 + \frac{\lambda}{4!}\,\varphi^4, \quad m^2 > 0, \quad V''(0) = -m^2. \tag{A.1}$$

Writing the Fourier modes as

$$\varphi_k(t) = \int d^3x\, e^{-i\mathbf{k}\cdot\mathbf{x}}\,\varphi(\mathbf{x}, t), \tag{A.2}$$

the linearized equation of motion around $\phi \equiv \langle\varphi\rangle = 0$ is

$$\left[\partial_t^2 + k^2 - m^2\right]\varphi_k = 0, \tag{A.3}$$

which implies the dispersion relation

$$\omega^2(k) = k^2 - m^2. \tag{A.4}$$

For modes with $k < m$, $\omega$ becomes purely imaginary and the general solution grows (or decays) exponentially:

$$\varphi_k(t) \propto e^{\pm\gamma(k)\,t}, \qquad \gamma(k) = \sqrt{m^2 - k^2}. \tag{A.5}$$

Hence the band of unstable (tachyonic) modes extends from $k = 0$ up to $k = m$, characteristic of spinodal decomposition. In the case of two coupled condensates one works with a complex two-component field (equivalently four real degrees of freedom). The fluctuation operator is then a $4 \times 4$ matrix; diagonalizing it yields dispersion relations which are in general complex. As before, the imaginary parts of the eigenfrequencies determine the growth rates of the unstable momentum modes.

## A.2  Parametric instability

The *parametric instability*, closely related to classical *parametric resonance*, is the second key mechanism in our study. In classical mechanics, parametric resonance occurs when, for example, an oscillating pendulum has its length (and hence its natural frequency) varied periodically: the periodic "pumping" of the frequency excites and amplifies specific oscillation modes. In quantum field theory a directly analogous effect arises whenever the background field oscillates in time. If the homogeneous condensate $\phi(t)$ induces a time-dependent mass term $m^2[\phi(t)]$, certain fluctuation modes of the field obey equations with periodically modulated frequency and can grow exponentially in the linear regime [40]. Similarly, one may consider explicit temporal modulation of couplings (e.g. the tunnel rate or interaction strength), which yields the same mathematical structure. By differentiating the fluctuation equation once more and eliminating the conjugate momentum, one finds, in the uncoupled-condensate limit, the Mathieu equation for the density perturbation [Eq. (25)].

# B    Mean-field approximation

This appendix contains the derivation of the mean field equations [Eq. (6)].

## B.1    Equations of motion

The classical equations of motion correspond to the zeroth order in a perturbative expansion of the full GPE [Eq. (3)] in terms of fluctuations. The equation for the $\Psi_j$ field becomes, in the phase-density representation [Eq. (4)]

$$\frac{i\hbar\dot{\bar{\rho}}_j}{2\sqrt{\bar{\rho}_j}}e^{i\bar{\phi}_j} - \hbar\dot{\bar{\phi}}_j\sqrt{\bar{\rho}_j}e^{i\bar{\phi}_j} = g\bar{\rho}_j\sqrt{\bar{\rho}_j}e^{i\bar{\phi}_j} - \hbar J\sqrt{\bar{\rho}_l}e^{i\bar{\phi}_l}, \tag{B.1}$$

with $j \neq l$. Multiplying both sides by $\sqrt{\bar{\rho}_j}$ leads to

$$\frac{i\hbar\dot{\bar{\rho}}_j}{2} - \hbar\bar{\rho}_j\dot{\bar{\phi}}_j = -\hbar J\sqrt{\bar{\rho}_j\bar{\rho}_l}[\cos(\bar{\phi}_l - \bar{\phi}_j) - i\sin(\bar{\phi}_l - \bar{\phi}_j)] + g\bar{\rho}_j^2. \tag{B.2}$$

Now, the equations of motion for the density and phase fields can be deduced by separating Eq. (B.2) into the imaginary and real parts, respectively, giving

$$\hbar\dot{\bar{\rho}}_j = 2J\hbar\sqrt{\bar{\rho}_j\bar{\rho}_l}\sin(\bar{\phi}_l - \bar{\phi}_j),$$
$$\hbar\dot{\bar{\phi}}_j = \hbar J\sqrt{\frac{\bar{\rho}_l}{\bar{\rho}_j}}\cos(\bar{\phi}_l - \bar{\phi}_j) - g\bar{\rho}_j. \tag{B.3}$$

Finally, the equations of motion can be expressed in terms of the relative degrees of freedom $\bar{z}$ and $\bar{\phi}$, leading to Eq. (6).

## B.2    An effective motion in a quartic potential

The equations of motion for the background fields $z, \phi$ [according to Eq. (6), where we drop the bar and the explicit dependence on time here for the sake of notational simplicity] can be recast in an effective equation of motion for the fractional population imbalance moving in an effective potential $V(z)$ as the following. To reduce the number of variables to a single one, we can use that energy (from now on we set $\hbar = 1$)

$$H = \frac{\mu z^2}{4} - J\sqrt{1 - z^2}\cos\phi \tag{B.4}$$

is conserved, i.e.,

$$H = H_0 \equiv H(t = 0) = \frac{\mu z_0^2}{4} - J\sqrt{1 - z_0^2}\cos(\phi_0). \tag{B.5}$$

As a result, the phase variable $\phi$ can be eliminated, and the resulting equation of motion for $z$ is given by the following equation

$$\ddot{z} = \left(2\mu H_0 - 4J^2\right)z - \frac{\mu^2}{2}z^3 \equiv -\frac{\partial V}{\partial z}. \tag{B.6}$$

In other words, the imbalance is effectively moving in a classical effective potential $V(z)$. The potential is obtained by integrating Eq. (B.6) and results in

$$V(z) = -\left(\mu H_0 - 2J^2\right)z^2 + \frac{\mu^2}{8}z^4. \tag{B.7}$$

The stationary point occurs when the initial condition $z_0$ aligns with the minimum of the effective potential, leading to Eq. (8).

In the case where the initial imbalance is very strong and $\Lambda \ll 1$, we have $H_0 \approx \mu/4$. In this scenario, Eq. (B.6) reduces to

$$\ddot{z} + \left( \frac{\mu^2}{2} - 4J^2 \right) z - \frac{\mu^2}{2} z^3 = 0. \tag{B.8}$$

The exact solution can be expressed in terms of Jacobi elliptic functions [86]

$$z(\tau) = \mathrm{dn}\left( \frac{2\tau}{\mu}, \frac{\Lambda}{4} \right). \tag{B.9}$$

### B.3 $\pi$-trapping

For a given initial phase $\phi_0 = \pi$ and population imbalance $z_0$, the two-mode dynamics depend sensitively on the interaction parameter $\Lambda$. It is instructive to contrast the $\pi$-trapped state of the two-mode Josephson junction with the inverted equilibrium state of the Kapitza pendulum. In the strict two-mode (0D) limit, the junction dynamics map onto a nonrigid pendulum whose effective length varies with the population imbalance $z$. For interaction strengths in the window $\Lambda_b < \Lambda < \Lambda_u$ (see subsect. 2.3.1 for a detailed discussion), intrinsic $\pi$-oscillations around the inverted angle ($\phi = \pi$) occur. By contrast, the Kapitza pendulum is a rigid pendulum whose upside-down equilibrium is unstable unless one applies a high-frequency vertical drive to its pivot: the rapid shaking reshapes the time-averaged potential to carve out a stabilizing well at $\theta = \pi$. Thus, while both systems support small oscillations around an inverted configuration, the Josephson $\pi$-oscillations stem purely from nonlinear interactions within a closed system, whereas the Kapitza oscillations require an external parametric drive.

## C Classical-statistical simulations

In this appendix, we summarize the main information about the implementation of the GPE numerical simulations presented in Sec. 4.

### C.1 Numerical implementation

The numerical technique of classical-statistical simulations (also known as truncated Wigner simulations) incorporates quantum fluctuations through stochastic initial conditions, while the time evolution is deterministic and defined by classical equations of motion (see e.g. [87,88]). The observables consist of quantum expectation values obtained via statistical averages across a large number of independent realizations [42]. In this work, we examine homogeneous scalar BECs, either at zero or finite temperature, defined on a spatial grid of length $L$ with periodic boundary conditions. For a single realization, we sample the initial field configuration from the Wigner distribution of the initial state, here taken to be the vacuum or thermal equilibrium state of the Bogoliubov quasiparticles, as

$$\Psi_j(0, x) = \sqrt{\rho_j(0)} e^{i\phi_j(0)} + \sqrt{\frac{\eta}{2}} \sum_{k \neq 0} \left[ \alpha_{k,j} u_{k,j}(x) - \alpha_{k,j}^* v_{k,j}^*(x) \right]. \tag{C.1}$$

The parameter $\eta$ is used to control the level of noise, with $\eta < 1$ reducing the vacuum noise below the average occupancy of half a particle per mode, i.e., the "quantum one-half." Explicit expressions for $u_{k,j}(x)$ and $v_{k,j}(x)$ are given by $u_{k,j}(x) = u_{k,j} e^{ikx/\hbar}/\sqrt{L}$ and $v_{k,j}(x) = v_{k,j} e^{ikx/\hbar}/\sqrt{L}$,

and

$$u_{k,j} = \sqrt{\frac{1}{2}\left(\frac{\xi_{k,j}}{\epsilon_{k,j}} + 1\right)}, \qquad v_{k,j} = \sqrt{\frac{1}{2}\left(\frac{\xi_{k,j}}{\epsilon_{k,j}} - 1\right)}, \tag{C.2}$$

are the solutions of the Bogoliubov-de-Gennes equations for a uniform system in a periodic box, with real coefficients $\epsilon_{k,j} = \sqrt{\epsilon_{k,0}(\epsilon_{k,0} + \rho_{0,j}g)}$, $\xi_{k,j} = \epsilon_{k,0} + \rho_{0,j}g$, and $\epsilon_{k,0} = (\hbar k)^2/(2m)$. It holds that $|u_{k,j}|^2 - |v_{k,j}|^2 = 1$. The quasiparticle amplitudes $\alpha_{k,j}$ in Eq. (C.2) are sampled as

$$\alpha_{k,j} = \sqrt{n_{BE,kj} + \frac{1}{2}} \frac{x_k + i y_k}{\sqrt{2}} \tag{C.3}$$

in order to mimic quantum fluctuations. In the last expression, $n_{BE,kj} = 1/(\exp(\epsilon_{k,j}/k_b T) - 1)$ is the Bose-Einstein distribution, and $x_k, y_k$ are normally distributed Gaussian random numbers with mean zero and unit variance:

$$\begin{aligned} \overline{\alpha_{p,j}} &= \overline{\alpha_{p,j}\alpha_{q,j}} = 0, \\ \overline{\alpha_{p,j}^* \alpha_{q,k}} &= (n_{BE,pj} + 1/2)\delta_{p,q}\delta_{j,k}. \end{aligned} \tag{C.4}$$

## C.2 Dimensionless units

We define a spatial discretization $a_G$ corresponding to the spacing of the numerical grid. We define the dimensionless time $\tilde{t}$, space $\tilde{x}$ and $j$-field $\tilde{\Psi}_j$ using the following transformations

$$z \to a_G \tilde{z}, \quad t \to \frac{\tilde{t}}{\omega_G}, \quad \Psi_j \to \frac{1}{\sqrt{a_G}}\tilde{\Psi}_j, \tag{C.5}$$

where $\omega_G = \hbar/(ma_G^2)$. Expressing Eq. (3) in $\tilde{\ }$ units results in

$$i\partial_{\tilde{t}}\tilde{\Psi}_1 = \left[-\frac{1}{2}\partial_{\tilde{x}}^2 + \tilde{g}|\tilde{\Psi}_1|^2\right]\tilde{\Psi}_1 - \tilde{J}\tilde{\Psi}_2, \tag{C.6}$$

with the dimensionless couplings

$$\tilde{g} = 2\frac{a_s}{a_G}\frac{\omega_\perp}{\omega_G}, \quad \tilde{J} = \frac{\omega_J}{\omega_G}, \tag{C.7}$$

using the fact that $g = 2\hbar a_s \omega_\perp$, where $a_s$ is the s-wave scattering length and $\omega_\perp$ the frequency of the radial confinement [7]. Explicitly, the left-hand side of Eq. (3) becomes

$$i\hbar\partial_t\Psi_1 = \frac{i\hbar\omega_G}{\sqrt{a_G}}\partial_{\tilde{t}}\tilde{\Psi}_1, \tag{C.8}$$

while the right-hand side becomes

$$\left[-\frac{\hbar^2}{2m}\partial_x^2 + g|\Psi_1|^2\right]\Psi_1 - \hbar J\Psi_2 = \left[-\frac{1}{2}\underbrace{\frac{\hbar^2}{ma_G^2}}_{\hbar\omega_G}\partial_{\tilde{x}}^2 + \frac{g}{a_G}|\tilde{\Psi}_1|^2\right]\frac{1}{\sqrt{a_G}}\tilde{\Psi}_1 - \hbar J\frac{1}{\sqrt{a_G}}\tilde{\Psi}_2. \tag{C.9}$$

Dividing both sides by $\hbar\omega_G/\sqrt{a_G}$ leads to Eq. (C.6).

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
