# Peer review of "Tachyonic and parametric instabilities in an extended bosonic Josephson junction"

_SciPost Physics_

## Round 1 · Referee Report · Anonymous (Referee 1) · 2025-2-3

Report

The authors study instabilities arising in two tunnel-coupled Bose-Einstein
condensates on the theoretical level and propose experimental realizations.
However, I think that some open questions need to be addressed:
The authors write: "After reviewing the stable mean-field dynamics,
we focus first on the effect of fluctuations on the linearized level,
destabilizing the state."
That sounds strange or even contradictory -- is the point here that a
spatially homogeneous mean-field solution is stable, while spatial
inhomogeneities can grow (similar to the origin of the CMB fluctuations)?
Then the authors use the phrase "tachyonic instability."
As far as I understand, this usually refers to the instability of a field
near the top of its effective potential.
As such, this instability should also occur at k=0 and may go away at
finite k (again similar to the origin of the CMB fluctuations).
However, what the authors find (if I understood correctly) is an
instability which is absent at k=0 but present at finite k?
Can the authors explain how this fits (if it does) to the top of an
effective potential?
It would also be nice to have a better (intuitive) understanding of
where the instability comes from (in the stationary case, the
parametric instability is more easy to grasp).
This would also help to justify publication in SciPost, I think.
Minor points: Eq.(1) is not normal ordered, i.e., singular.
In a footnote, the authors write: "We have indicated this regime with
the gray shaded area in Fig. 2." but I could not find it.

Recommendation

Ask for major revision

  • validity: -
  • significance: -
  • originality: -
  • clarity: -
  • formatting: -
  • grammar: -

Author:  Laura Batini  on 2025-09-29  [id 5871]

(in reply to Report 1 on 2025-02-03)

We would like to thank the Referee for the insightful feedback, which helped us to significantly improve our manuscript. Below, we give detailed answers to each of the referee's questions and points of critique and mention the corresponding modifications in the new version.

Referee: The authors write: "After reviewing the stable mean-field dynamics, we focus first on the effect of fluctuations on the linearized level, destabilizing the state." That sounds strange or even contradictory -- is the point here that a spatially homogeneous mean-field solution is stable, while spatial inhomogeneities can grow (similar to the origin of the CMB fluctuations)?

Our response: We agree with the Referee’s interpretation that the spatially homogeneous mean-field solution is indeed stable against uniform perturbations, while spatially varying (finite-$k$) fluctuations can grow. In the CMB, tiny variations propagate as smooth sound waves, while in our case those same differences explode in amplitude thanks to both tachyonic and parametric instabilities. To avoid any confusion, we have revised the manuscript by rephrasing the original sentence
After reviewing the stable mean-field dynamics, we focus first on the effect of fluctuations on the linearized level, destabilizing the state.

as follows:

First, we recapitulate the spatially uniform mean-field dynamics, which is stable. Next we analyze small, spatially dependent fluctuations. Specifically, first we expand the dynamics up to first order in space-time dependent perturbations. We find that bands of momentum modes with nonzero wavevector are exponentially growing and quantify the corresponding growth rate.

Referee: Then the authors use the phrase "tachyonic instability." As far as I understand, this usually refers to the instability of a field near the top of its effective potential. As such, this instability should also occur at k=0 and may go away at finite k (again similar to the origin of the CMB fluctuations). However, what the authors find (if I understood correctly) is an instability which is absent at k=0 but present at finite k? Can the authors explain how this fits (if it does) to the top of an effective potential?

Our response: We appreciate the Referee’s point that “tachyonic” typically means a negative mass‑squared already at $k=0$. In the textbook single‑field potential

$$ V(\varphi) = -\frac{m^2}{2}\,\varphi^2 + \frac{\lambda}{4!}\,\varphi^4, \quad m^2>0, \quad V''(0)=-m^2. $$
this indeed yields $\omega^2(k) = k^2 -m^2$, and the unstable band running from $k=0$ to $k=m$ corresponds to the momenta where the dispersion relation develops a nonvanishing imaginary part. By contrast, in our two‑condensate model, in the case of two coupled condensates one works with a complex two-component field (equivalently four real degrees of freedom). The fluctuation operator is then a 4 × 4 matrix; diagonalizing it yields dispersion relations that are, in general, complex. As discussed above, the imaginary parts of the eigenfrequencies determine the growth rates of the unstable momentum modes. The resulting instability bands and their growth rate are illustrated in Fig. 4 in the manuscript, and the corresponding analytical dispersion relation given in Eq.~(17). As shown in Fig.~4, in our case the unstable modes are not centered at $k=0$, but rather around two finite, nonzero momenta.

Referee: It would also be nice to have a better (intuitive) understanding of where the instability comes from. (in the stationary case, the parametric instability is easier to grasp).

Our response: The instability reflects pair emission from the higher-energy condensate. When at least one quasiparticle tunnels to the lower-energy well, the released potential energy $\Delta \mu$ is converted into kinetic energy of a back-to-back pair (total momentum $\sim$ 0), producing two peaks in the momentum distribution. Two processes satisfy energy–momentum conservation: (i) single tunneling, where one quasiparticle remains while its partner tunnels (energy $\Delta \mu$), and (ii) pair tunneling, where both tunnel together (energy $2 \Delta \mu$). See Ref. Twin Atom Beams, R. Bucker et al. Nature Physics, 7, 608-611 (2011) for an experimental analogue in twin-atom beams.

Referee: Minor points. Our response: We thank the referee to point out these deficiencies and have addressed the requested changes: - Normal-ordered Eq. (1) to remove the singular self-interaction term. - Corrected the footnote by changing the reference from “Fig. 2” to “Fig. 3.”

---

## Round 1 · Referee Report · Anonymous (Referee 2) · 2025-5-1

Strengths

The authors investigate the dynamics and decay of quantum phase coherence in tunnel-coupled elongated Bose-Einstein condensates (BECs). The authors focus particularly on the behavior of the so-called π-mode, i.e. a self-trapped oscillatory state, and its instability due to quantum fluctuations. The study combines linearized analysis of early-time dynamics with numerical simulations in the Truncated Wigner approximation to explore both the onset and evolution of nonequilibrium instabilities, including tachyonic and parametric resonance modes. The paper concludes with a discussion of experimental parameters relevant for observing these phenomena.
The work addresses a timely topic in quantum many-body dynamics and the analysis is well motivated, combining analytic and numerical approaches to gain insight into the dynamics beyond the mean-field regime.

Weaknesses

In particular the distinction between tachyonic and parametric resonance mechanisms, is valuable. 1.While the physics is rich, the manuscript would benefit from an introduction to the key physical concepts like “tachyonic” and “parametric” instabilities. 2. The manuscript would be strengthened by a more explicit comparison to prior theoretical and experimental studies of Josephson dynamics and self-trapping in BECs. For instance, how does this analysis extend or differ from earlier Truncated Wigner treatments of coupled condensates? E.g in Phys. Rev. B 106, 075426 the same type of Mathieu equations are discussed to describe the higher order resonances. As for the self-trapping part, how this study extends the Smerzi two mode model? 3. Regarding the quantum-fluctuation-induced instability of the π-mode it would be helpful to more clearly distinguish which features are inaccessible within mean-field theory. Is this instability somehow related to the Kapitza pendulum physics? 4. It’s not very clear why the primary instability is associated to the excitation of a pair of particles from the first condensate, to the second. Do they mean excitations made by bound particles? 5. The discussion of experimental parameters is a valuable aspect of the paper. However, it would be helpful to quantify more explicitly what detection signatures (e.g., momentum distributions, coherence fringes) would signal the presence of the described instabilities in a real experiment.

Report

I recommend revisions prior to publication. The results are interesting and merit publication, but some clarifications and additional references to previous similar studies would strengthen the manuscript.

Requested changes

See above

Recommendation

Ask for minor revision

  • validity: good
  • significance: high
  • originality: good
  • clarity: good
  • formatting: excellent
  • grammar: excellent

Author:  Laura Batini  on 2025-09-29  [id 5872]

(in reply to Report 2 on 2025-05-01)
Category:
remark
answer to question
correction

We thank the Referee for recognizing that our work addresses a timely topic in quantum many-body dynamics and the analysis is well motivated, combining analytic and numerical approaches to gain insight into the dynamics beyond the mean-field regime. Their insightful and constructive suggestions helped us to significantly improve our manuscript. We have changed the manuscript according to their remarks and, in the following, we give a detailed answer to each of the referees' questions and points of critique.

Referee: While the physics is rich, the manuscript would benefit from an introduction to the key physical concepts like "tachyonic" and "parametric" instabilities.

Our response: Following the referee’s recommendation, we have added in the new version more information on the concepts of tachyonic and parametric instabilities. The paragraphs below have been added in a new appendix.

In quantum field theory out of equilibrium, instabilities are characterized by exponential amplification of quantum fluctuations. These behaviors typically manifest in two-point correlation functions. This mechanism leads to rapid amplification of low-energy fluctuations. Exponentially growing correlations can arise via different physical mechanisms. In the following, we introduce the two different types which arise in our model presented in Sec. 2. The tachyonic instability is often referred to as spinodal decomposition, or simply spinodal instability. Consider a single‐component real scalar field $\varphi$ with potential

$$ V(\varphi) = -\frac{m^2}{2}\,\varphi^2 + \frac{\lambda}{4!}\,\varphi^4, \quad m^2>0, \quad V''(0)=-m^2. $$
Writing the Fourier modes as
$$ \varphi_k(t) = \int d^3x\;e^{-i\mathbf{k}\cdot\mathbf{x}}\;\varphi(\mathbf{x},t), $$
the linearized equation of motion around $\phi\equiv\langle\varphi\rangle=0$ is
$$ \bigl[\partial_t^2 + k^2 - m^2\bigr]\,\varphi_k = 0, $$
which implies the dispersion relation
$$ \omega^2(k) = k^2 - m^2. $$
For modes with $k<m$, $\omega$ becomes purely imaginary and the general solution grows (or decays) exponentially:
$$ \varphi_k(t) \propto e^{\pm \gamma(k)\,t}, \qquad \gamma(k) = \sqrt{\,m^2 - k^2\,}. $$
Hence the band of unstable (tachyonic) modes extends from $k=0$ up to $k=m$, characteristic of spinodal decomposition. In the case of two coupled condensates one works with a complex two‐component field (equivalently four real degrees of freedom). The fluctuation operator is then a 4x4 matrix; diagonalizing it yields dispersion relations which are in general complex. As before, the imaginary parts of the eigenfrequencies determine the growth rates of the unstable momentum modes.

The parametric instability — closely related to classical parametric resonance— is the second key mechanism in our study. In classical mechanics, parametric resonance occurs when, for example, an oscillating pendulum has its length (and hence its natural frequency) varied periodically: the periodic pumping of the frequency excites and amplifies specific oscillation modes. In quantum field theory, a directly analogous effect arises whenever the background field oscillates in time. If the homogeneous condensate $\phi(t)$ induces a time‐dependent mass term $m^2[\phi(t)]$, certain fluctuation modes of the field obey equations with periodically modulated frequency and can grow exponentially in the linear regime [see Berges, Nonequilibrium Quantum Fields: From Cold Atoms to Cosmology, arXiv:1503.02907]. Similarly, one may consider explicit temporal modulation of couplings (e.g., the tunnel rate or interaction strength), which yields the same mathematical structure. By differentiating the fluctuation equation (Eq. (16) in the main text) once more and eliminating the conjugate momentum, one finds—in the uncoupled‐condensate limit—the Mathieu equation for the density perturbation $\delta\rho_j$:

$$ \frac{d^2\delta\rho_j}{ds^2} + \bigl(A - 2\,q\,\cos(2s)\bigr)\,\delta\rho_j = 0, $$
with $s\equiv\omega t$.

Referee: The manuscript would be strengthened by a more explicit comparison to prior theoretical and experimental studies of Josephson dynamics and self-trapping in BECs. For instance, how does this analysis extend or differ from earlier Truncated Wigner treatments of coupled condensates? E.g in Phys. Rev. B 106, 075426 the same type of Mathieu equations are discussed to describe the higher order resonances. As for the self-trapping part, how this study extends the Smerzi two mode model?

Our response: We understand the Referee to be pointing out that our original manuscript could have provided a more clearly structured comparison with prior works. Therefore, we substantially revised the introduction, gave it a clear structure, and added more detail comparing our present work with previous theoretical and experimental studies.

More specifically, in response to the Referee’s remark on Ref. Phys. Rev. B 106, 075426, which likewise employs Mathieu equations, we note that whereas that work enforces parametric growth via explicit modulation of the tunneling coupling, in our system the oscillating mean field $\phi(t)$ provides the periodic drive naturally, producing identical instability bands. Consequently, the Mathieu equation governing our parametric resonance coincides with Eq. (13) of Phys. Rev. B 106, 075426.

Referee: Regarding the quantum-fluctuation-induced instability of the $\pi$-mode it would be helpful to more clearly distinguish which features are inaccessible within mean-field theory. Is this instability somehow related to the Kapitza pendulum physics?

Our response: We thank the referee for this remark. We would like to clarify the difference between the physics of the Kapitza pendulum and the one of the $\pi$-trapped state. The Kapitza pendulum is a rigid pendulum whose upside-down equilibrium is unstable unless one applies a high-frequency vertical drive to its pivot: the rapid driving creates a time-averaged potential which develops a stabilizing well at $\theta=\pi$.
By contrast, the bosonic Josephson junction dynamics map onto a nonrigid pendulum (whose effective length varies with the population imbalance z), and the inverted equilibrium point is stable in mean-field. Therefore, the $\pi$-trapped state of the two-mode Josephson junction and the inverted equilibrium of the Kapitza pendulum both support small oscillations around an inverted configuration, but the Josephson $\pi$-oscillations stem purely from nonlinear interactions within a closed system, whereas the Kapitza oscillations are stabilized by the external parametric drive.

The referee remarks: It's not very clear why the primary instability is associated to the excitation of a pair of particles from the first condensate, to the second. Do they mean excitations made by bound particles?

Our response: Energy–momentum conservation demands that these excitations be produced with opposite momenta, and only when one member of the pair tunnels out of the higher-energy condensate is there excess energy available to appear as kinetic energy. A simple kinematic argument then explains the two observed peaks: Single tunneling, when one quasiparticle remains in the original well while its partner tunnels to the lower-energy well, converting the released energy into finite momentum of the pair; Pair tunneling: both quasiparticles tunnel together into the lower-energy condensate, again sharing the excess energy as kinetic motion. In both cases, momentum conservation guarantees back-to-back emission (total momentum zero), and no extra symmetry constraints arise since a quasiparticle localized in either well is itself a superposition of symmetric and antisymmetric states. We have rewritten Subsec. 3.2 to emphasize that allowing quasiparticle tunneling leads inevitably to the emission of correlated pairs into either well.

Referee: However, it would be helpful to quantify more explicitly what detection signatures (e.g., momentum distributions, coherence fringes) would signal the presence of the described instabilities in a real experiment.

Our response:

  • We rewrote and extended the section regarding experimental feasibility (now Sec. 5). We now highlight the different measurement schemes in a more systematic way. In addition we included some preliminary data (two new figures) from pilot experiments exploring a very similar setting: Macroscopic Quantum Self Trapping (MQST). This allows to illustrate how we can reconstruct the time evolution of both the phase and the imbalance from the data, and how we can measure the longitudinal momentum distribution which should give information about the decay channels of the $\pi$-state.
  • We also checked finite-temperature initial states and found that the primary instability bands remain robust at realistic $T$ (Subsec. 4.1).
  • Additional clarifications: We would also like to point out two further aspects, which we have now included in the revised manuscript. First, while our focus in Sec. 3 is on the primary tachyonic and parametric instabilities, our simulations (Sec. 4, Figs. 9–10) also reveal secondary instability peaks at higher momenta. These arise from nonlinear scattering of the primary excitations. Although experimentally more demanding to observe, they could in principle be accessed with higher statistics and correlation analysis. Second, in Subsec. 4.1, we present finite-temperature simulations which confirm that the primary instability bands remain robust under realistic initial conditions, underscoring the experimental feasibility of observing these effects with current cold-atom technology.

---

## Round 2 · Referee Report · Anonymous (Referee 1) · 2025-11-27

Report

The authors have improved their manuscript in response to the referee reports.
To be honest, I am still not fully happy regarding the term "tachyonic instability"
but I understand that the authors use this term to distinguish that mechanism
from the parametric instability (which is clearly different).
However, this slight misgiving is not a reason to prevent publication.

Recommendation

Publish (easily meets expectations and criteria for this Journal; among top 50%)

---

## Round 2 · Author Response

We thank the Referees for their insightful feedback and for the constructive suggestions on our manuscript. We have carefully considered all the comments and addressed them point by point. A detailed response is provided below each referee report. In this revised version of the manuscript, we have clarified the points mentioned in the referee reports.

---

## Round 2 · List of Changes

• Eq. (1) is now normal-ordered to remove the singular self-interaction term.
  • We corrected the footnote with the correct figure reference (Fig. 3).
  • We substantially revised the introduction with a clearer structure and expanded the comparison with previous theoretical and experimental work.
  • We clarified the role of momentum-dependent instabilities vs. uniform modes and provided additional intuitive understanding of the instabilities.
  • We revised the discussion of the primary instability mechanism, adding a new appendix introducing mode details on the tachyonic and parametric instabilities.
  • We added the comparison between the $\pi$-state instability with the Kapitza pendulum.
  • We extended and systematized the discussion of experimental feasibility (now Sec. 5). We added two new figures (Fig. 12 and Fig. 13) with preliminary experimental data (Sec. 5).

---

## Editorial Decision

accepted_in_target_journal